# Every animal matters! Evaluating the selectivity of a Mediterranean bottom trawl fishery from a species community perspective

**Andrea Petetta**[1,2]*, **Bent Herrmann**[3,4,5], **Daniel Li Veli**[2], **Massimo Virgili**[2], **Rocco De Marco**[2], **Alessandro Lucchetti**[2]

**1** Department of Biological, Geological and Environmental Sciences (BIGEA), University of Bologna (UNIBO), Bologna, Italy, **2** Institute for Biological Resources and Marine Biotechnologies (IRBIM), National Research Council (CNR), Ancona, Italy, **3** The Arctic University of Norway UIT, Tromsø, Norway, **4** SINTEF Ocean, Fishing Gear Technology, Trondheim, Norway, **5** Technical University of Denmark, National Institute of Aquatic Resources, Hirtshals, Denmark

* andrea.petetta@irbim.cnr.it

**Data Availability Statement:** All relevant data are within the paper and its Supporting Information files.

## Abstract

Bottom trawl fisheries often catch several species simultaneously. However, most studies addressing the catch performance and selectivity of a specific trawl focus on a few commercially important or most vulnerable species requiring management measures. By contrast, the present study considers the multispecies nature of Mediterranean bottom trawl fisheries through a holistic approach that accounts for the full species community in the catches. Specifically, we evaluated and compared the catch performance of the two codends allowed for this fishery, made of 40 mm square (SM40) and 50 mm diamond (DM50) meshes. Results showed that 50 and 80% of the catch in weight and count numbers, respectively, consisted of species without commercial value, demonstrating that large proportions of the catch are not considered when using the existing approach to evaluate the ecological impact of the fishing activity. Significant differences in catch profiles between the two codends were observed, especially for two commercial flatfish species, *Arnoglossus laterna* and *Citharus linguatula*, with larger contributions in the SM40. Further, the SM40 codend had a significantly higher retention, compared to DM50 codend, for specific sizes of *Merluccius merluccius* and *Mullus barbatus*. The outcomes of the study can be useful for the Mediterranean bottom trawl fisheries management.

## 1. Introduction

Most of the demersal trawl fisheries operating in the Mediterranean Sea simultaneously target different species sharing the same grounds at the same time [1]. Besides them, there are several other species caught more or less frequently which can both provide additional income to fishers, or be discarded because of little or no commercial value.

The catch of unwanted species is one of the main issues in bottom trawl fisheries' management, since it contributes, together with the catch of undersized or damaged specimens of

**Funding:** This study was conducted with the contribution of the LIFE Financial Instrument of the European Community, Life Delfi Project – Dolphin Experience: Lowering Fishing Interactions (LIFE18NAT/IT/000942; https://lifedelfi.eu/). The funders had no role in study design, data collection and analysis, decision to publish, or preparation of the manuscript.

**Competing interests:** The authors have declared that no competing interests exist.

commercial species, to the high discard rates reported throughout the basin (20–65% of the total catch, according to Tsagarakis et al. [2]). Most of the discards might not survive because they are damaged in the capture process, sometimes hauled up from the fishing depth too quickly, or thrown back too late. Since these fish, crustaceans, shellfish etc. are part of an ecosystem, their removal affects the entire food chain [3]. In addition, the rejecting practises of dead or dying organisms does not result in any economic advantage, as catches cannot be sold for human consumption and will not benefit fishing activities in future years [4].

The minimization of this wasteful practice of discarding is one of the pillars of the Common Fisheries Policy (CFP), which, in the Mediterranean basin, enforced several specific frameworks of fisheries management [5, 6]. First, the CFP established the landing obligation (LO) of all the catches of species subjected to the minimum conservation reference size (MCRS; [5]). Other measures concern the establishment of fisheries restricted areas [7], initiatives to facilitate control by authorities and to provide incentives to fishers to improve compliance [8], and the enforcement of technical limitations on fishing gears. In bottom trawl fisheries, these limitations concern the mesh size and geometry at the trawl codend level. In fact, the codend is considered to be the main part of the net where the fish escapement process takes place [9, 10]. Currently, the codend allowed for the Mediterranean has to be constructed with 40 mm (full mesh) square meshes (SM40, hereafter); an alternative legal codend, only with a duly justified request from the shipowner, is the 50 mm (full mesh) diamond mesh (DM50, hereafter) codend [6]. However, both these codends are unable to avoid a considerable capture of unwanted species and immature and undersized fish, thus being inefficient at ensuring a sustainable harvest for the majority of the fish stocks [1].

Most of the scientific studies investigating the catch patterns, size and species selectivity of trawls usually focus on a few species. These are often the most economically important species [11] or most vulnerable species requiring urgent management measures [12]. On the contrary, the multispecies nature of Mediterranean bottom trawl fisheries is rarely addressed, despite the growing need for ecosystem-based fishery management for conserving biodiversity [13]. However, the overall ecological impact of using a specific gear or technical modification cannot be determined if significantly important fractions of the catch are neglected, which may cause an underestimation of the unaccounted fishing mortality [14]. Accordingly, there is a need for a more holistic approach that considers the whole species community in the bottom trawl catches. The present study, therefore, aims at using a methodology that takes into account all the animals being caught intentionally or unintentionally and being both landed or discarded. This methodology was applied to sea trials carried out in the North-western Adriatic Sea (FAO Geographical Sub Area 17). The objective was to assess the selectivity of the two codends (SM40 and DM50) from a species community perspective, and evaluate if there were changes in the catch profiles, in terms of species composition and dominance, when shifting from one codend to another. Moreover, a comparison of the catch efficiency at size between SM40 and DM50 was performed on the most abundant commercial species.

## 2. Materials and methods

### 2.1. Sea trials and data collection

The sea trials were conducted 13–16 nautical miles off the coast of Senigallia (central Italy; Fig 1) from 16th to 25th February 2022 on board R/V "G. Dallaporta" (810 kW at 1650 rpm, Length Over All 35.30 m, Gross Tonnage 285 GT).

The gear used in the experiments was a "Volantina" net often employed by commercial trawlers of the area. It is an asymmetric 2-panels net (with the upper panel shorter than the lower panel) entirely made of knotless polyamide (PA) netting (see Sala and Lucchetti [15] for

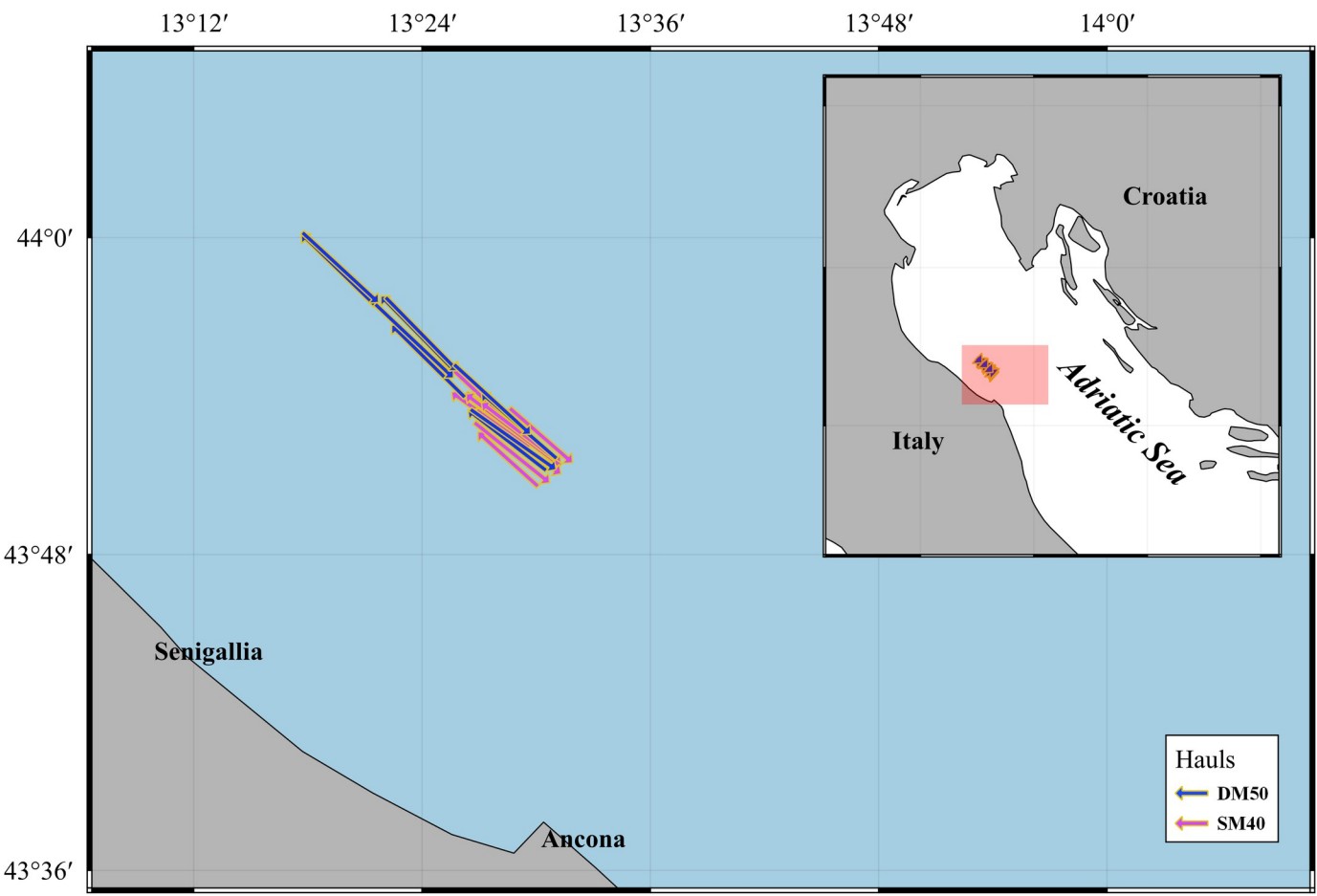

**Fig 1. Map of the hauls carried out during the sea trials, distinguished by codend type (SM40: 40 mm square mesh codend; DM50: 50 mm diamond mesh codend).** The arrows indicate the towing direction.

trawl design). The length from the wing tips to the codend was approximately 60 m, with 600 meshes in the top panel at the footrope level. The sweeps and bridles were 80 and 50 meters long, respectively. A single pair of otter boards (163x100 cm, 270 kg each) was used to maintain the horizontal net opening.

Both codends tested (SM40 and DM50) were made of the same netting twine and twine thickness (knotless PA, 3880 RTEX). Their mean measured mesh size, obtained with an

**Table 1. Technical features of the two codends tested.** SM40: 40 mm square mesh codend; DM50: 50 mm diamond mesh codend.

|  | SM40 | DM50 |
|---|---|---|
| Codend length (m) | 4.8 | 4.8 |
| Nr. Meshes in codend circumference | 140 | 220 |
| Mesh configuration | Square | Diamond |
| Nominal mesh size (mm) | 40 | 50 |
| Measured mesh size (mm ± SD) | 38.11 ± 0.94 | 50.26 ± 0.81 |
| Netting twine | PA (Knotless) | PA (Knotless) |
| Twine thickness (RTEX) | 3880 | 3880 |

OMEGA gauge [16] at 50 N while the netting was wet, was 38.1 mm for SM40 and 50.3 mm for DM50. The number of meshes at codend circumference was 140 for SM40 and 220 for DM50 (Table 1), to obtain a similar circumference between codends during fishing, thus avoiding different effects on selectivity. In fact, following the considerations and calculations made by Sala et al. [17] on the expected mesh openness during fishing in square and diamond configurations, respectively, the expected circumference of both codends was around 2.7 meters.

Each of the two codends was mounted on the same trawl. 19 valid hauls were performed: the first 9 hauls with SM40 and the last 10 with DM50. All hauls were carried out in daylight at a mean depth (SD) of 55.5 meters (1.2), with a standardized towing duration of around 60 minutes (59.2 ± 1.9). The average towing speed was 3.0 knots (range 2.8–3.2 knots). The horizontal net opening (19.5 ± 0.2 meters) was monitored using acoustic sensors (SIMRAD, Norway).

After each haul, the catch was sorted following the commercial procedures. The fishers on deck divided the total catch into a discarded and a landed fraction. Then the researchers on board divided the catch of each fraction by species, by identifying all taxa to the lowest possible level. For each species within each fraction, the total weight was recorded and the individuals were counted. The counts were performed without sub-sampling except for the large catches of swimming crab (*Liocarcinus depurator*) obtained in each haul, for which the count was conducted on a randomly selected subsample, and the subsample coefficient was determined. Furthermore, individual length measurements (total length for fish and mantle length for cephalopods) were taken to the lowest 0.5 cm for the most abundant commercial species caught in the sea trials. These were: European hake (*Merluccius merluccius*), red mullet (*Mullus barbatus*), Atlantic mackerel (*Scomber scombrus*), Mediterranean scaldfish (*Arnoglossus laterna*), spotted flounder (*Citharus linguatula*), broadtail shortfin squid (*Illex coindetii*) and tub gurnard (*Chelidonichthys lucerna*). The measurements were recorded without sub-sampling except for a large catch obtained for *A. laterna* in one haul, where measurements were conducted on a randomly selected subsample.

## 2.2. Data analysis

The statistical software SELNET [18] was used to analyze the catch data. Data were treated as unpaired [19], since the shift from one codend to another was not done after each haul and therefore we could not treat them as paired hauls.

**2.2.1. Catch dominance analysis.** Catch dominance curves are often used to quantify information about the relative species abundances for a given sample. The catch dominance analysis was here performed to evaluate, for each species, its relative abundance in both the total catch, the discarded catch and the landed catch. The aim was to assess if the proportion of each species in each fraction was significantly different between the two codends tested.

Usually, the dominance curves are based on the ranking of species in a sample in decreasing order of their abundance [20]. In the present study, a fixed rank was assigned to each single species caught in the sea trials, by including it into one of the following 4 categories: 1) 'Target species', i.e. the main commercial species targeted by the Italian Adriatic bottom trawl fishery in 20–100 meters of depth [21]; 2) 'Bycatch species of commercial value', i.e. additional species with a commercial value, that are landed; 3) 'Species of no commercial value', i.e. those species usually discarded by the fishers; 4) 'Protected species', i.e. those species included in EU regulations and International lists (e.g. EU Habitat directive, IUCN red list).

Since our intent was to estimate, on average, the performance of the two codends in the fishery, the catch dominance curves were averaged over hauls. They were then estimated, in

both number of individuals ($dn_i$) and weight ($dw_i$), for each codend and for each fraction of the catch (total, discarded, landed) by using the following equations [11, 22]:

$$dn_i = \sum_{j=1}^{h} \left\{ \frac{\frac{n_{ij}}{q_{ij}}}{\sum_{i=1}^{s} \left\{ \frac{n_{ij}}{q_{ij}} \right\}} \right\} \tag{1}$$

$$dw_i = \sum_{j=1}^{h} \left\{ \frac{\rho_{ij} \times \frac{n_{ij}}{q_{ij}}}{\sum_{i=1}^{s} \left\{ \rho_{ij} \times \frac{n_{ij}}{q_{ij}} \right\}} \right\} \tag{2}$$

where $j$ represents the haul and $i$ is the species rank previously defined. $n_{ij}$ is the number of individuals of the species $i$ being counted in the subsample in haul $j$. $q_{ij}$ represents the subsampling ratio, i.e. the counted subsample of species $i$ in haul $j$. Parameter $\rho$ is the average weight of species $i$ in haul $j$ in a given fraction of the catch, and it is obtained from the total weight and number of individuals. $S$ is the total number of species considered, whereas $h$ is the total number of hauls conducted with the specific codend.

The cumulative dominance curves were then estimated, in both number of individuals ($Dn_I$) and weight ($Dw_I$), to better represent species dominance patterns, as follows:

$$Dn_I = \sum_{j=1}^{h} \left\{ \frac{\sum_{i=1}^{I} \frac{n_{ij}}{q_{ij}}}{\sum_{i=1}^{s} \left\{ \frac{n_{ij}}{q_{ij}} \right\}} \right\} \text{ with } 1 \leq I \leq S \tag{3}$$

$$Dw_I = \sum_{j=1}^{h} \left\{ \frac{\sum_{i=1}^{I} \rho_{ij} \times \frac{n_{ij}}{q_{ij}}}{\sum_{i=1}^{s} \left\{ \rho_{ij} \times \frac{n_{ij}}{q_{ij}} \right\}} \right\} \text{ with } 1 \leq I \leq S \tag{4}$$

where $I$ is the species rank summed up in the nominator.

The Efron percentile 95% confidence intervals (CIs; [23]) were used to provide the uncertainty of the values of dominance patterns obtained following the procedure described in Herrmann et al. [24]. This procedure enables estimation of the uncertainty around the dominance values at species level induced by the limited sample sizes at single haul without having to make any prior assumptions regarding the distributions in the hauls [24].

Furthermore, the difference $\Delta d$ in species dominance $d$ in the SM40 ($x$) and DM50 ($y$) codends was estimated by:

$$\Delta d = d_y - d_x \tag{5}$$

By applying the technique described in Herrmann et al. [24], the CIs for Eq 5 were obtained based on separate bootstrap populations for $d_x$ and $d_y$. The significance was detected by inspecting if the CIs contained the value 0.0. If the 0.0 value was within the CIs, no significant difference was detected.

**2.2.2. Catch comparison and catch ratio analysis.** The length-frequency distributions, obtained from the most abundant commercial species mentioned above, were used to perform a catch comparison and catch ratio analysis. The aim was to investigate the size-dependent effect on the catch efficiency of each species by changing the codend.

For each species independently, we assessed the relative length-dependent catch comparison rate ($CC_l$) of shifting from one codend to another, by using Eq 6 [19]:

$$CC_l = \frac{\sum_{j=1}^{ht} \left\{ \frac{nt_{lj}}{qt_j} \right\}}{\sum_{j=1}^{hb} \left\{ \frac{nb_{lj}}{qb_j} \right\} + \sum_{j=1}^{ht} \left\{ \frac{nt_{lj}}{qt_j} \right\}} \tag{6}$$

where $nb_{lj}$ and $nt_{lj}$ are the number of fish of length $l$ of a given species retained in haul $j$ by the baseline codend ($b$, i.e. SM40 codend) and test codend ($t$, i.e. DM50 codend), respectively. Parameters $qb_j$ and $qt_j$ are the subsampling ratios, i.e. the ratios of the measured to the total number of individuals retained by the baseline and the test codend, respectively. Parameters $hb$ and $ht$ represent the total number of hauls conducted with baseline and test codend, respectively.

We estimated the catch comparison rate $CC(l,v)$ experimentally expressed by Eq 6, by minimizing the Expression 7 (maximum likelihood estimation):

$$-\sum_l \left\{ \sum_{j=1}^{hb} \left\{ \frac{nb_{lj}}{qb_j} \times ln[CC(l, v)] \right\} + \sum_{j=1}^{ht} \left\{ \frac{nt_{lj}}{qt_j} \times ln[1.0 - CC(l, v)] \right\} \right\} \tag{7}$$

where the outer summation is over the length classes $l$ and the inner summation is over the hauls $ht$ and $hb$ in the experimental dataset. The $v$ parameter describes the catch comparison curve defined by $CC(l,v)$. The experimental $CC_l$ was modelled by the function $CC(l,v)$:

$$CC(l, v) = \frac{exp[f(l, v_0, \ldots, v_k)]}{1 + exp[f(l, v_0, \ldots, v_k)]} \tag{8}$$

where $f$ is a polynomial of order $k$ with coefficients $v_0$ to $v_k$, such that $v = (v_0, \ldots, v_k)$. $f$ was considered up to an order of 4. Leaving out one or more of the parameters $v_0 \ldots v_4$ yielded 31 additional candidate models for the catch comparison function $CC(l,v)$. We estimated the catch comparison rate, among these models, by using the multi-model inference to obtain a combined model [19, 25]. We based the ability of the combined model to describe the experimental data on the $p$-value, calculated based on the ratio between the model deviance and the degrees of freedom (DOF; [19, 26]). A $p$-value $> 0.05$ indicates suitable fit statistics for the combined model to describe the experimental data sufficiently well. With poor fit statistics ($p$-value $< 0.05$ and deviance/DOF $>> 1$), the residuals were inspected to determine whether the results were due to structural problems when modelling the experimental data, or to overdispersion in the data [26].

$CC(l,v)$ quantifies the probability that a fish of length $l$ is retained by the DM50 codend, provided that it is retained in one of the two codends (DM50 or SM40). Since the number of valid hauls conducted with the two codends was different (10 hauls with DM50, 9 hauls with SM40), the same probability, for a fish with a given length $l$, of being retained by either gear will be at $CC(l) = 0.526$ (i.e. the ratio 10 to 19).

The results of $CC(l,v)$ do not provide a direct relative value of the catch efficiency between the test and the baseline codends. Therefore, we used the catch ratio $CR(l,v)$, since it provides

such direct comparison and can be easily derived from *CC(l,v)* following the equation:

$$CR(l, \mathbf{v}) = \frac{hb \times CC(l, v)}{ht \times [1 - CC(l, v)]} \tag{9}$$

In this case, *CR(l,v)* = 1.0 means that the catch efficiency of both codends is equal, while *CR (l,v)* = 0.25 indicates that the test net is catching only 25% of the fish of length *l* compared to the baseline net.

We then estimated the 95% *CIs* for both the catch comparison and catch ratio curves by using a double bootstrapping method with 1000 bootstrap repetitions. By using this approach, following the description given in Lomeli [27], both within and between haul variations were taken into account.

**2.2.3. Exploitation pattern indicator analysis.** We applied the exploitation pattern indicators for catch comparison to summarize the relative performance of the two codends tested. The indicators were adopted from Bonanomi et al. [28] and Veiga-Malta et al. [29] on the seven species previously selected for the catch comparison and catch ratio analysis. Among them, only *M. merluccius*, *M. barbatus* and *S. scombrus* are subjected to a minimum conservation reference size (MCRS of 20, 11 and 15 cm, respectively; [30]; therefore, individuals below this size were considered as discards. Regarding *A. laterna*, the 12 cm length class was considered as a reference size between the individuals discarded (<12 cm) and landed (≥12 cm), according to common fishers' practises. All the individuals of *C. linguatula*, *I. coindetii* and *C. lucerna* caught were landed, thus no reference sizes were available for those species.

For the species with a reference size, the average percentage of individuals below and above this size retained by the test codend (DM50) compared to the baseline codend (SM40), in terms of number of individuals (*nP-*, *nP+*) were estimated as follows:

$$nP- = \frac{hb \times \sum_{j=1}^{ht} \sum_{l < MCRS} \left\{ \frac{nT_{jl}}{qT_j} \right\}}{ht \times \sum_{j=1}^{hb} \sum_{l < MCRS} \left\{ \frac{nB_{jl}}{qB_j} \right\}} \tag{10}$$

$$nP+ = \frac{hb \times \sum_{j=1}^{ht} \sum_{l \geq MCRS} \left\{ \frac{nT_{jl}}{qT_j} \right\}}{ht \times \sum_{j=1}^{hb} \sum_{l \geq MCRS} \left\{ \frac{nB_{jl}}{qB_j} \right\}} \tag{11}$$

where $nT_{jl}$ and $qT_j$ represent the estimation made for the test codend, and $nB_{jl}$ and $qB_j$ the estimation made for the baseline codend. The summations of *j* and *l* in (10) and (11) are over the hauls *ht* and *hb*, and length classes *l*, respectively. An indicator value of 100% means that the test codend caught an equal number of individuals below (*nP-*) and above (*nP+*) the MCRS, respectively, compared to the baseline codend. Indicator values of 50% and 150% mean that the test codend caught 50% less and 50% more individuals (below or above MCRS) than the baseline codend, respectively.

For those species not subject to a MCRS or without a reference size, the mean percentage of all individuals retained by the test codend compared to the baseline codend was estimated, in

number of individuals (*nP*), as follows:

$$nP = \frac{hb \times \sum_{j=1}^{ht} \left\{ \sum_l \frac{nT_{jl}}{qT_j} \right\}}{ht \times \sum_{j=1}^{hb} \left\{ \sum_l \frac{nB_{jl}}{qB_j} \right\}} \tag{12}$$

A value of 100% means that the test codend catches the same total number (*nP*) of the species analysed as the standard codend.

Discard ratios were then estimated for each gear in terms of number for both the test (*nDRatioT*) and baseline (*nDRatioB*) codends, as follows:

$$nDRatioT = 100 \times \frac{\sum_{j=1}^{ht} \left\{ \sum_{l < MCRS} \frac{nT_{jl}}{qT_j} \right\}}{\sum_{j=1}^{ht} \left\{ \sum_l \frac{nT_{jl}}{qT_j} \right\}} \tag{13}$$

$$nDRatioB = 100 \times \frac{\sum_{j=1}^{hb} \left\{ \sum_{l < MCRS} \frac{nB_{jl}}{qB_j} \right\}}{\sum_{j=1}^{hb} \left\{ \sum_l \frac{nB_{jl}}{qB_j} \right\}} \tag{14}$$

A discard ratio of 0% would imply that no discard was produced. Again, the 95% *CIs* for each indicator were estimated using the double bootstrap method described above.

### 2.3. Ethic statements

The fishing trials carried out on board the research vessel have been authorised by the Italian coastguard. The only protected species caught during sea trials is *Alosa fallax*, included in the list of Annexes II and V of animals requiring close protection under the Habitats Directive. No other authorization or ethics board approval was required. No information on animal welfare or on steps taken to mitigate fish suffering and methods of sacrifice is provided since the animals were not exposed to any additional stress other than that involved in commercial fishing practices. This article does not contain any studies with human participants performed by any of the authors.

## 3. Results

### 3.1. Catch dominance analysis

A total of 68 species belonging to 9 higher taxa (Osteichthyes and Condrichthyes; Crustacea Decapoda; Mollusca Cephalopoda, Bivalvia and Gastropoda; Echinodermata; Polychaeta; Ascidiacea) were caught during the sea trials (Table 2). Only 4 fish species were included in the 'Target species' category: European hake, red mullet, monkfish and tub gurnard. 27 species were classified as 'Bycatch species of commercial value'; they included 16 fish species (13 bony fishes and 3 elasmobranches), 7 cephalopod species and 4 crustacean species. 37 species were classified as 'Species of no commercial value'. Most of them were fish (21 species), followed by crustaceans (6 species), echinoderms (5 species), molluscs (3 species), polychaetes (1 species) and ascidian (1 species). Only 1 protected species (the twait shad, *Alosa fallax*, listed in

**Table 2. Assigned ranking (S) of the animal species, divided by category, caught during the sea trials.**

| Target species | | Bycatch species of commercial value | | Species of no commercial value | | Protected species | |
|---|---|---|---|---|---|---|---|
| S1 | *Merluccius merluccius* | S5 | *Trisopterus minutus capelanus* | S32 | *Liocarcinus depurator* | S68 | *Alosa fallax* |
| S2 | *Mullus barbatus* | S6 | *Scomber scombrus* | S33 | *Medorippe lanata* | | |
| S3 | *Lophius* spp | S7 | *Trachurus mediterraneus* | S34 | *Goneplax rhomboides* | | |
| S4 | *Chelidonichthys lucerna* | S8 | *Alloteuthis media* | S35 | *Dardanus arrosor* | | |
| | | S9 | *Loligo vulgaris* | S36 | *Maja squinado* | | |
| | | S10 | *Sepia officinalis* | S37 | *Solenocera membranacea* | | |
| | | S11 | *Eledone* spp. | S38 | *Sardina pilchardus* | | |
| | | S12 | *Illex coindetii* | S39 | *Engraulis encrasicolus* | | |
| | | S13 | *Octopus vulgaris* | S40 | *Sardinella aurita* | | |
| | | S14 | *Sepia elegans* | S41 | *Spicara maena* | | |
| | | S15 | *Raja asterias* | S42 | *Boops boops* | | |
| | | S16 | *Raja clavata* | S43 | *Blennius ocellaris* | | |
| | | S17 | *Squalus acanthias* | S44 | *Conger conger* | | |
| | | S18 | *Citharus linguatula* | S45 | *Eutrigla gurnardus* | | |
| | | S19 | *Arnoglossus laterna* | S46 | *Gobius niger* | | |
| | | S20 | *Solea solea* | S47 | *Lepidotrigla cavillone* | | |
| | | S21 | *Scophthalmus rhombus* | S48 | *Lesuerigobius friesii* | | |
| | | S22 | *Nephrops norvegicus* | S49 | *Microchirus variegatus* | | |
| | | S23 | *Melicertus kerathurus* | S50 | *Pagellus acarne* | | |
| | | S24 | *Squilla mantis* | S51 | *Pagellus bogaraveo* | | |
| | | S25 | *Parapenaeus longirostris* | S52 | *Pagellus erythrinus* | | |
| | | S26 | *Sparus aurata* | S53 | *Pagrus pagrus* | | |
| | | S27 | *Uranoscopus scaber* | S54 | *Callionymus* spp. | | |
| | | S28 | *Trachinus draco* | S55 | *Cepola macrophthalma* | | |
| | | S29 | *Merlangius merlangus* | S56 | *Scorpaena notata* | | |
| | | S30 | *Pomatomus saltator* | S57 | *Serranus hepatus* | | |
| | | S31 | *Zeus faber* | S58 | *Sphyraena sphyraena* | | |
| | | | | S59 | *Phallusia mamillata* | | |
| | | | | S60 | *Ocnus planci* | | |
| | | | | S61 | *Marthasterias glacialis* | | |
| | | | | S62 | *Astropecten irregularis* | | |
| | | | | S63 | *Aphrodite aculeata* | | |
| | | | | S64 | *Armina tigrina* | | |
| | | | | S65 | *Ostrea edulis* | | |
| | | | | S66 | *Mytilus galloprovincialis* | | |
| | | | | S67 | *Stichopus regalis* | | |

Annexes II and V of the Habitats Directive as requiring close protection; [31]) was caught during the cruise. This classification was specifically related to the fishers' choice in a precise spatiotemporal context. Therefore, it does not fully represent neither all the Mediterranean bottom trawl fisheries, where some species, here classified as commercial, could be always discarded elsewhere and vice versa due to local consumers' preferences, nor the Adriatic fishery, where some species can have a market value only in a specific season.

S1 Table shows the catch dominance percentages, in both number of individuals and weight, of each species in each codend tested (SM40 and DM50). The first 31 species often had both a landed and a discarded fraction, since some animals were rejected because they were

too few or too small, damaged, with little commercial value in that specific area and season or below the MCRS. The latter 37 species were always discarded.

Fig 2 represents, for each codend, the cumulative species dominance (in percentages), in both number of individuals and weight, of the three catch fractions: total catch, discarded catch and landed catch. The cumulative curve of the total catch shows that, in the DM50 codend, the targeted species (S1-S4) and the species with a commercial value (S5-S31) cover, on average, less than 15% in number of individuals and less than 50% in weight of all the species caught. This means that the species with no commercial value represent the largest proportion of the total catch. The results are slightly different for the SM40, where the proportions of the first 31 species reach, on average, the 20% (in number of individuals) and the 50% (in weight) of all the catches. In both codends, there is a dramatic curve increase at species 32, corresponding to the swimming crab (*Liocarcinus depurator*), which brings the curve up to almost 90% considering both individuals and weight (Fig 2; see also S1 Table for the proportion of each species in the catch).

Regarding the discarded fraction of the catch, the proportion of the first 17 species (S1-S17) is low in both codends, but there is a clear increase in S18 and S19 (corresponding to two flatfish species, *C. linguatula* and *A. laterna*). This increase is more marked in SM40 than in DM50, in both number of individuals and weight. Again, in both codends *L. depurator* makes the curve rise from less than 10% to almost 90% when considering individuals and to around 80% when considering weight (Fig 2 and S1 Table). Regarding the landed fraction of the catch, the cumulative curve reaches 100% within the first 31 species. A significant curve increase is observed, in SM40 (number of individuals), in S18 and S19 (i.e. the flatfish species above mentioned). This increase is not discernible in the corresponding curve of DM50 (Fig 2).

Fig 3 shows the delta plots resulting from the comparisons between the cumulative dominance curves obtained with the two codends for both the total, discarded and landed fractions of the catch. The left column takes into account the number of individuals. The delta plot reveals a significant difference, in the total catch, from species S1 to S31 i.e. all the species of commercial interest, since both the upper and lower *CIs* are always below the 0.0 line which expresses an equal proportion between SM40 and DM50. A larger proportion, for these species, was in fact captured by the SM40. The same trend is observed in the discarded catch, meaning that SM40 also produced a larger proportion of discards for these species than the DM50, especially for the two flatfish species (S18, S19). Regarding the landed fraction of the catch, the DM50 produced a significantly larger proportion of the species from S10 to S17, which include commercial cephalopods and elasmobranches. No significant differences are observed in the proportion of the other species.

The right column takes into account the weight. In the total catch, a significant difference between the two codends is observed for the species S2 (*M. barbatus*), where the upper *CI* is below the 0.0 line, highlighting a lower proportion of this species in DM50 compared to SM40 catches. The same trend is observed, in the discarded catch, for species S1 to S31 i.e. all the species of a commercial value, especially from S18 to S31. Other barely significant differences (the upper *CI* almost reaches the 0.0 line) are observed in the left end of the curves of both total and discarded fractions from species S45 to S58 i.e. 'Species of no commercial value', whose proportion in the DM50 codend is slightly less than in the SM40 codend. Concerning the landed catch, the only significant difference concerns S2, whose proportion in DM50 is significantly less than in the SM40 (Fig 3, bottom left).

Fig 4 shows the delta plots resulting from the comparison of the catch dominance curves, for the total, discarded and landed fractions of the catch. This figure provides a detailed insight at each single species level. The plots of total and discarded fractions show, in particular, a significant difference between codends for the European hake (S1) and the two flatfish species

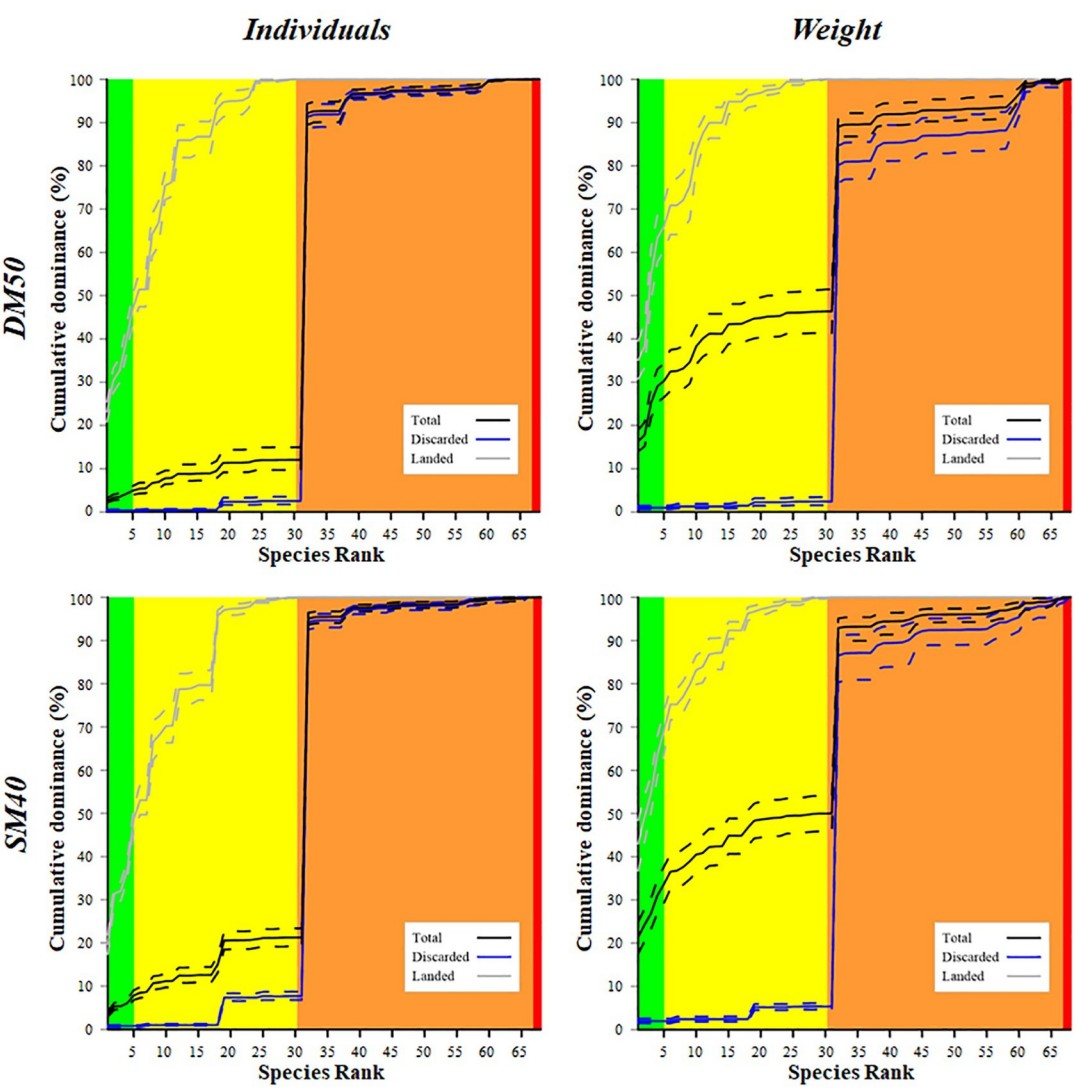

**Fig 2. Cumulative species dominance in the catch of the 50 mm diamond mesh codend (DM50, above) and 40 mm square mesh codend (SM40, below).** The curves (solid lines) with 95% CIs (dotted lines) represent the cumulative species dominance for the catches in both number of individuals (left) and weight (right). The green, yellow, orange and red areas represent the target species, the bycatch species of commercial value, the species of no commercial value and the protected species, respectively.

(S18, S19), indicating a larger proportion of their dominance in the SM40 catches than in the DM50 catches in both number of individuals (Fig 4, left column) and weight (Fig 4, right column). The delta plot of the total catch also shows a significantly larger proportion of red mullet (S2) in SM40 than in DM50 catches, only in weight; on the contrary, a significantly larger proportion of monkfish (S3) is present in DM50 catches, in both number of individuals and weight. Also, *L. depurator* (S32) had a significantly larger proportion in the total catch of DM50 compared to SM40, only in number of individuals. Other differences detected at species level for these two fractions are barely significant. The delta plot of the landed fraction, in both number of individuals and weight, shows in particular that a larger proportion of the target species *M. merluccius* (S1), *Lophius* spp. (S3), the cephalopods *S. officinalis* (S10) and *Eledone* spp. (S11), is present in the DM50 catches than in the SM40 catches. On the contrary, a larger

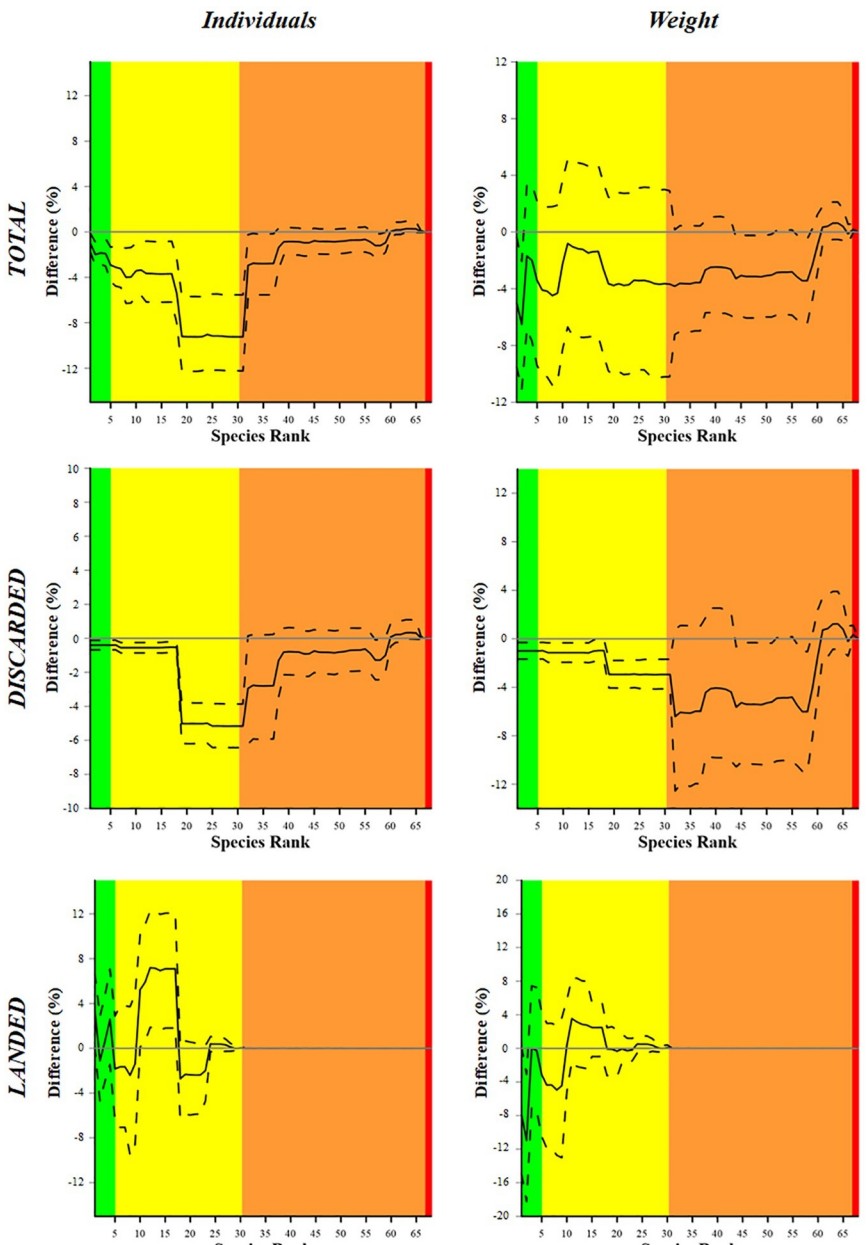

**Fig 3. Delta plots resulting from the comparison of the cumulative dominance curves between the two codends tested (DM50, SM40) in both number of individuals (left column) and weight (right column).** The curves (solid lines) with 95% CIs (dotted lines) are relative to the Total (top), discarded (middle) and landed (bottom) fractions. The 0 grey horizontal line represents an equal proportion between the two codends. The green, yellow, orange and red areas represent the target species, the bycatch species of commercial value, the species of no commercial value and the protected species, respectively.

proportion of *M. barbatus* (S2), *T. minutus capelanus* (S5) and *C. linguatula* (S18) is present in SM40 catches than in DM50 catches.

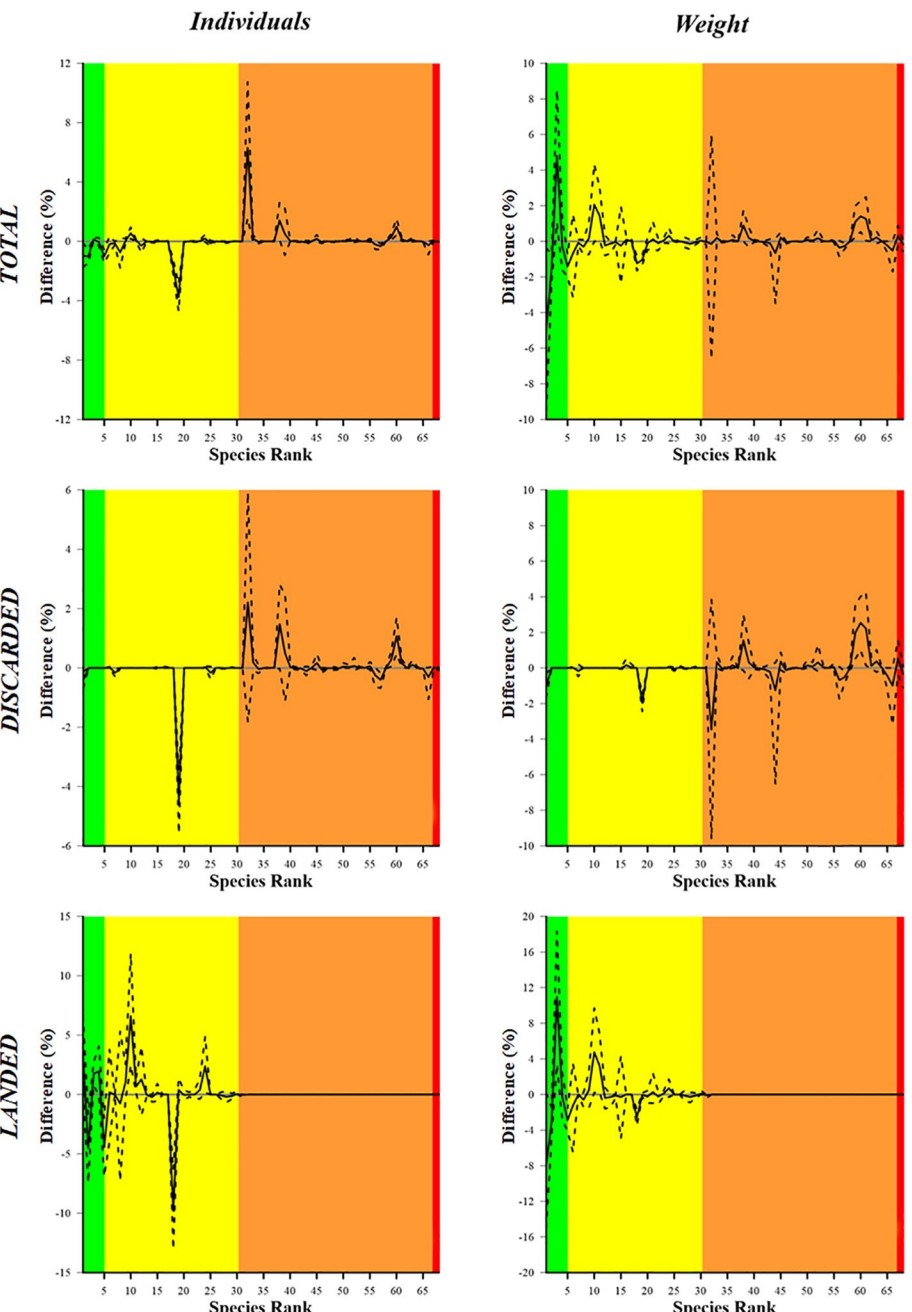

**Fig 4. Delta plots resulting from the comparison of the species catch dominance between the DM50 and the SM40, in both number of individuals (left column) and weight (right column).** The curves (solid lines) with 95% CIs (dotted lines) are relative to the Total (top), discarded (middle) and landed (bottom) fractions. The 0 grey horizontal line represents an equal proportion between the two codends. The green, yellow, orange and red areas represent the target species, the bycatch species of commercial value, the species of no commercial value and the protected species, respectively.

**Table 3. Number of individuals of the most abundant commercial species measured in each haul, selected for the catch comparison analyses.** No sub-samplings were performed except in one case, where the value of the subsampling coefficient is in brackets. SM40: 40 mm square mesh codend; DM50: 50 mm diamond mesh codend.

|  | Haul | *Merluccius merluccius* | *Mullus barbatus* | *Scomber scombrus* | *Arnoglossus laterna* | *Citharus linguatula* | *Illex coindetii* | *Chelidonichthys lucerna* |
|---|---|---|---|---|---|---|---|---|
| **SM40** | 1 | 97 | 60 | 4 | 133 | 49 | 28 | 22 |
|  | 2 | 90 | 29 | 9 | 122 | 60 | 38 | 18 |
|  | 3 | 95 | 40 | 15 | 143 | 61 | 33 | 19 |
|  | 4 | 53 | 37 | 23 | 133 | 37 | 14 | 14 |
|  | 5 | 55 | 31 | 10 | 79 | 37 | 6 | 8 |
|  | 6 | 65 | 20 | 2 | 72 (0.43) | 68 | 28 | 12 |
|  | 7 | 85 | 27 | 1 | 107 | 52 | 23 | 15 |
|  | 8 | 50 | 47 | 44 | 108 | 28 | 34 | 11 |
|  | 9 | 93 | 40 | 14 | 86 | 62 | 34 | 17 |
| **DM50** | 10 | 41 | 6 | 19 | 44 | 10 | 15 | 12 |
|  | 11 | 57 | 13 | 6 | 35 | 22 | 10 | 14 |
|  | 12 | 47 | 18 | 21 | 51 | 14 | 25 | 8 |
|  | 13 | 67 | 23 | 6 | 43 | 23 | 31 | 12 |
|  | 14 | 75 | 22 | 13 | 41 | 13 | 22 | 19 |
|  | 15 | 55 | 9 | 0 | 22 | 11 | 8 | 19 |
|  | 16 | 61 | 18 | 18 | 42 | 9 | 32 | 26 |
|  | 17 | 48 | 10 | 1 | 28 | 8 | 23 | 13 |
|  | 18 | 43 | 11 | 5 | 31 | 12 | 11 | 9 |
|  | 19 | 48 | 17 | 5 | 45 | 6 | 21 | 6 |

## 3.2. Catch comparison analysis

Table 3 reports the number of individuals measured in each haul for the 7 species selected for the catch comparison analysis (*M. merluccius*, *M. barbatus*, *S. scombrus*, *A. laterna*, *C. linguatula*, *I. coindetii*, *C. lucerna*).

Table 4 reports the fit statistics of the combined model. The *p*-value were < 0.05 for *A. laterna* ($p = 0.0071$), *I. coindetii* ($p = 0.0179$) and *C. lucerna* ($p = 0.026$). These low *p*-values were assumed to be due to overdispersion in the experimental rates rather than a lack of fit, since the visual inspection of the modelled catch comparison curves against the experimental rates for these species did not indicate any length-dependent patterns in the deviations (see Figs 6 and 7).

Figs 5–7 show the catch comparison and catch ratio results obtained by comparing the two codends (DM50 VS SM40) for the species selected. Regarding the European hake (Fig 5, top) a difference is observed from the 12 cm to the 26.5 cm length class, for which both the catch comparison and catch ratio curves display a significantly lower catch efficiency of the DM50 codend compared to the SM40 codend. A significantly lower retention of DM50 compared to SM40 is also observed for red mullet (Fig 5, middle) from 11 to 14.5 cm. This difference is well discernible not only from the catch comparison and catch ratio curves, but

**Table 4. Fit statistics of the combined model used in the catch comparison between the 40 mm square mesh codend and 50 mm diamond mesh codend.** DOF: degrees of freedom.

|  | *Merluccius merluccius* | *Mullus barbatus* | *Scomber scombrus* | *Arnoglossus laterna* | *Citharus linguatula* | *Illex coindetii* | *Chelidonichthys lucerna* |
|---|---|---|---|---|---|---|---|
| *p*-value | 0.1408 | 0.0846 | 0.0739 | 0.0071 | 0.1741 | 0.0179 | 0.026 |
| Deviance | 59.7 | 22.98 | 17.02 | 34.54 | 21.12 | 40.72 | 36.62 |
| DOF | 49 | 15 | 10 | 17 | 16 | 24 | 22 |

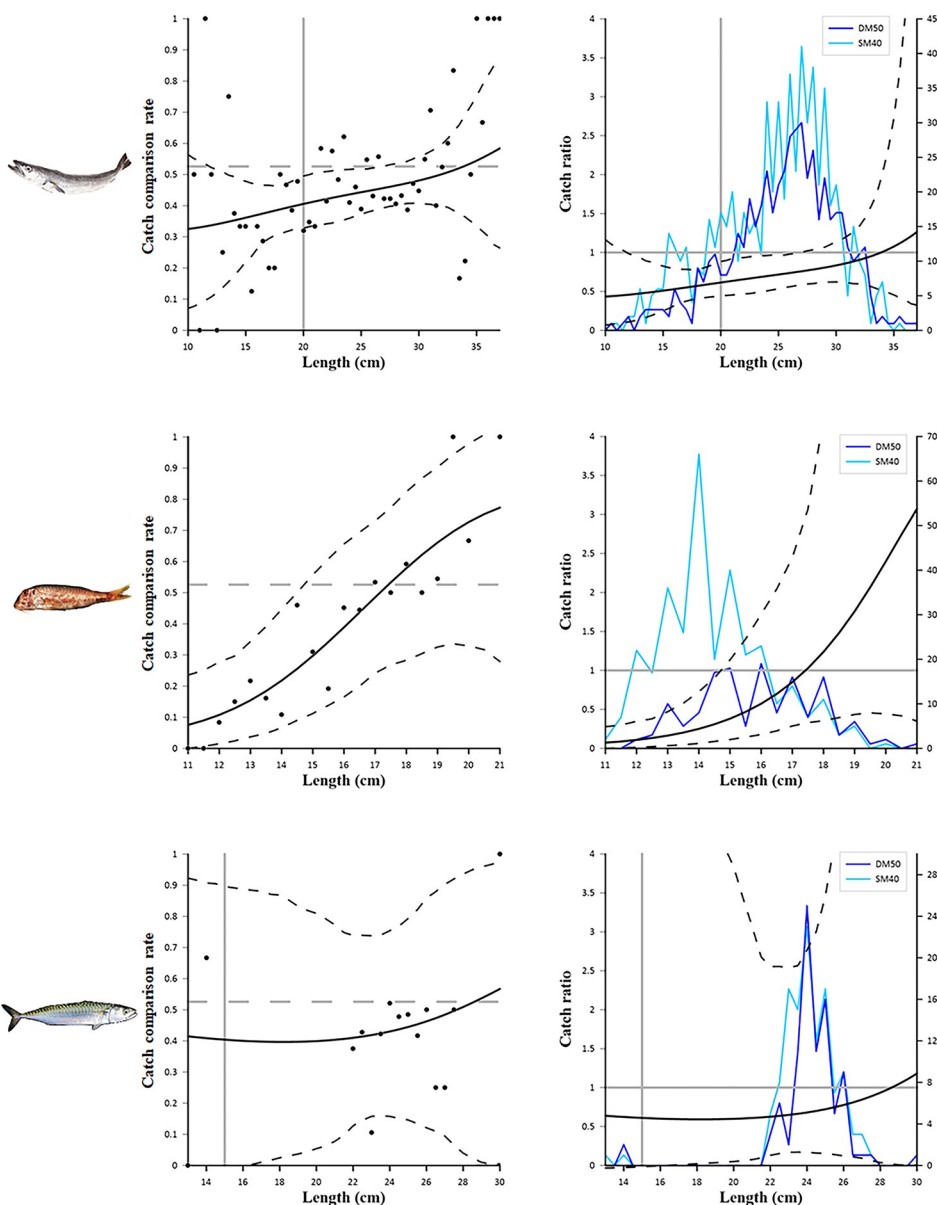

**Fig 5. Results of the catch comparison analyses obtained for *Merluccius merluccius* (top), *Mullus barbatus* (middle) and *Scomber scombrus* (bottom).** The graphs on the left show the modelled catch comparison rate (black line) with 95% *CI* (black stippled curves); the black circles represent the experimental rate; the grey horizontal line at 0.526 represents the point at which both configurations have equal catch rates; the grey vertical lines represent the MCRS of the species. The graphs on the right show the catch ratio (black line) with 95% *CI* (black stippled curves); the blue lines represent the length frequency distributions obtained with the two codends tested (DM50 and SM40); the grey horizontal line at 1.0 represents the point at which both configurations have equal catch rates; the grey vertical line represents the MCRS of the species.

also from the length frequency distributions obtained. No individuals below 11 cm, which represents the MCRS of the species, were caught in the sea trials. Regarding the Atlantic mackerel (Fig 5, bottom), there are no significant differences between the catch efficiency of the two codends, since the *CI*s of both the catch ratio and catch comparison curves over-lapped, for the full length range measured, the horizontal 0.526 line representing equal catch rates. The low number of individuals caught during the sea trials are reflected in the

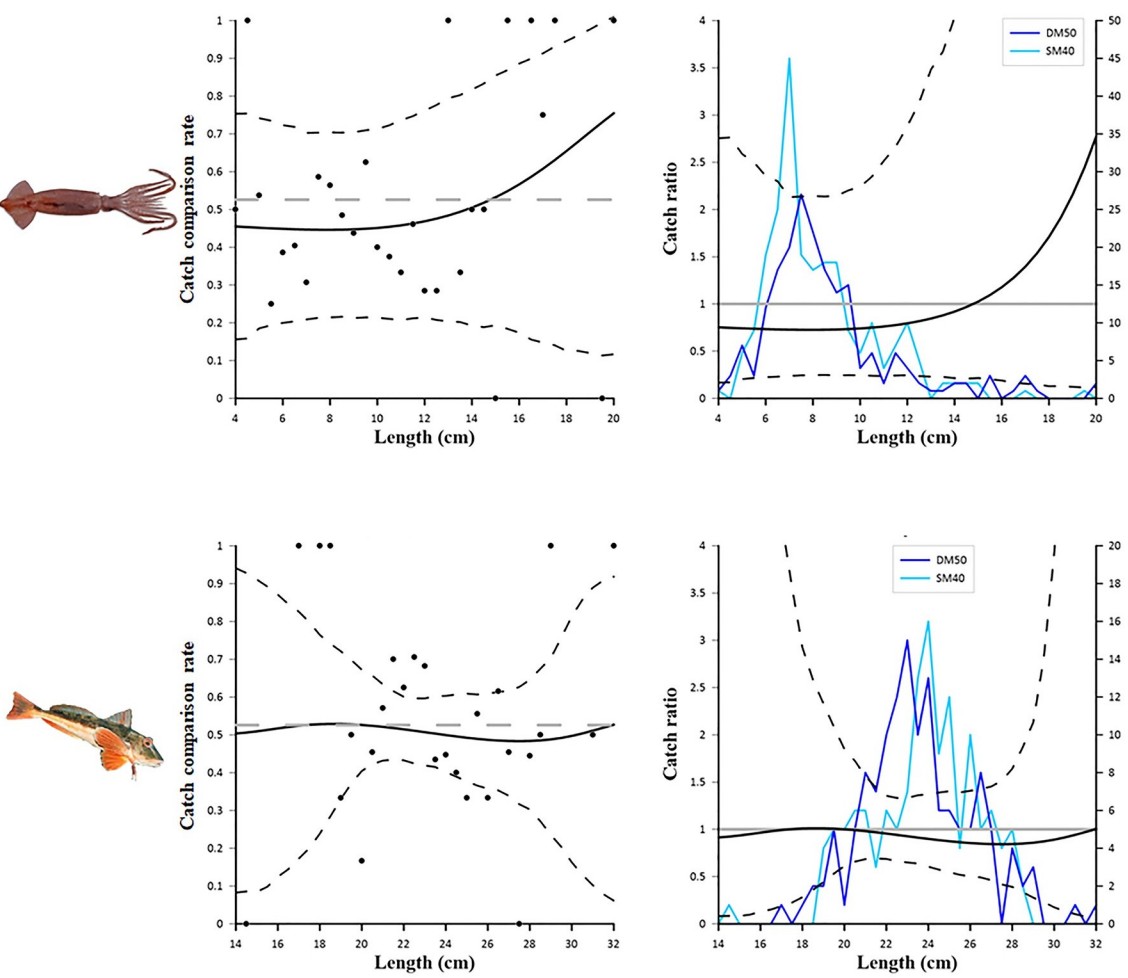

**Fig 7. Results of the catch comparison analyses obtained for *Illex coindetii* (top) and *Chelidonichthys lucerna* (bottom).** The graphs on the left show the modelled catch comparison rate (black line) with 95% *CI* (black stippled curves); the black circles represent the experimental rate; the grey horizontal line at 0.526 represents the point at which both configurations have equal catch rates. The graphs on the right show the catch ratio (black line) with 95% *CI* (black stippled curves); the blue lines represent the length frequency distributions obtained with the two codends tested (DM50 and SM40); the grey horizontal line at 1.0 represents the point at which both configurations have equal catch rates.

wide *CI*s of the curves. Also, very few individuals below 15 cm (i.e. the MCRS of the species) were caught in the sea trials.

Fig 6 shows the catch comparison results for the two flatfish species. The curves obtained for *A. laterna* clearly indicate a significantly lower catch efficiency of DM50, compared to SM40, in the 6.5 to 13.5 cm length range. In fact, the length frequency distributions show a clear decrease, in DM50, of the catch of individuals within this length range, when compared to SM40. The same trend is observed for *C. linguatula*, for which the DM50 codend display a significantly lower retention than the SM40 codend in the 9–16 cm length range. Again, the difference between the catches of the two codends is evident not only from the catch comparison and catch ratio curves but also from the length frequency distributions.

Fig 7 shows the results obtained for two important commercial species, the squid *I. coindetii* and the fish *C. lucerna*. In both cases, no significant differences in the catch efficiency of the two codends were detected. The wide *CI*s of both catch comparison and catch ratio curves and

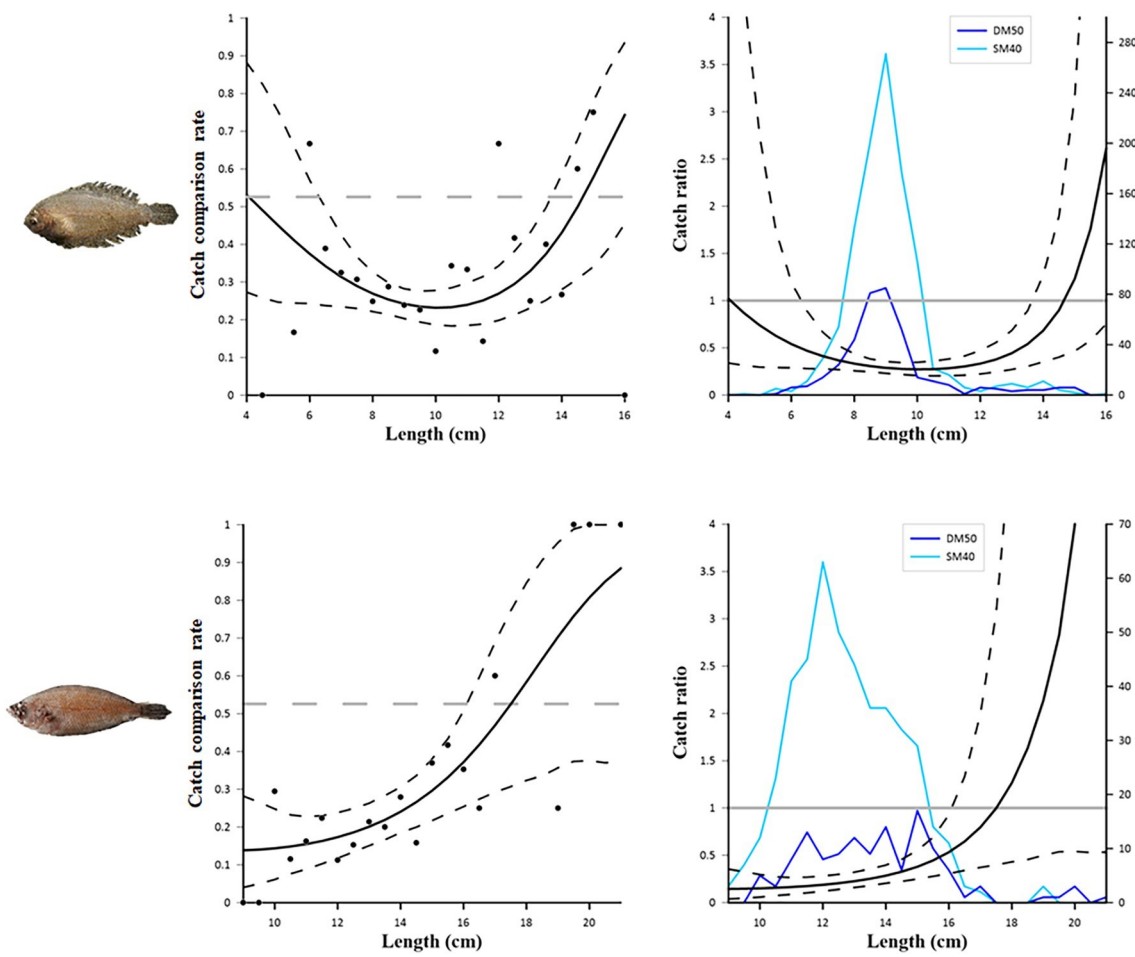

**Fig 6. Results of the catch comparison analyses obtained for *Arnoglossus laterna* (top) and *Citharus linguatula* (bottom).** The graphs on the left show the modelled catch comparison rate (black line) with 95% *CI* (black stippled curves); the black circles represent the experimental rate; the grey horizontal line at 0.526 represents the point at which both configurations have equal catch rates. The graphs on the right show the catch ratio (black line) with 95% *CI* (black stippled curves); the blue lines represent the length frequency distributions obtained with the two codends tested (DM50 and SM40); the grey horizontal line at 1.0 represents the point at which both configurations have equal catch rates.

the high dispersion in the experimental rates are probably due to a relatively low number of individuals caught during the sea trials.

## 3.3. Exploitation pattern indicator analysis

Table 5 shows the results obtained from the exploitation pattern indicator analysis. Regarding the European hake, the test codend (DM50) caught on average, in number of individuals, 50% (*nP*-; *CIs* 32.4–73.0%) less undersized individuals ($< 20$ cm) than the baseline codend (SM40). A slight but significant difference was found also for the individuals above the 20 cm MCRS (*nP*+) since the average percentage retained by DM50 was 76% (*CIs*: 60.0–97.9%) compared to SM40. The mean discard ratio, in number of individuals, obtained with DM50 (13.1%) was less than the mean discard ratio obtained with SM40 (18.6%), although the difference was not statistically significant (the *CIs* of the two values overlapped; Table 5).

The *nP*+ estimated for red mullet revealed that DM50 caught, on average, 44.4% of individuals above the 11 cm MCRS compared with SM40. However, the *CIs* of this indicator contained 100 (*CIs*: 7.8–118.3%), reflecting a lack of significant differences between the two

**Table 5. Values of the exploitation pattern indicators (in average percentages with 95% confidence intervals) for the species selected for the catch comparison analysis.** They represent the number of individuals i.e. nP (total), nP- (below the reference size), nP+ (above the reference size) retained by the test codend (DM50) compared to the baseline codend (SM40), and the resulting discard ratios estimated for both the test (nDRatioT) and baseline (nDRatioB) codends.

| Species | Indicator | Mean % (95% CI) |
|---|---|---|
| *Merluccius merluccius* | *nP-* | 50.3 (32.4–73.0) |
| **MCRS = 20 cm** | *nP+* | 76.2 (60.0–97.9) |
| | *nDRatioT* | 13.1 (8.9–17.9) |
| | *nDRatioB* | 18.6 (14.7–23.0) |
| *Mullus barbatus* | *nP-* | * |
| **MCRS = 11 cm** | *nP+* | 44.4 (7.8–118.3) |
| *Scomber scombrus* | *nP-* | * |
| **MCRS = 15 cm** | *nP+* | 69.0 (22.8–152.2) |
| *Arnoglossus laterna* | *nP-* | 33.7 (8.1–94.0) |
| **Reference size = 12 cm** | *nP+* | 79.1 (10.9–236.4) |
| | *nDRatioT* | 91.1 (86.7–94.5) |
| | *nDRatioB* | 96.0 (94.2–97.5) |
| *Citharus linguatula* | *nP* | 25.8 (18.7–35.2) |
| *Chelidonichthys lucerna* | *nP* | 92.0 (66.2–124.6) |
| *Illex coindetii* | *nP* | 83.2 (17.4–235.3) |

*Very few or no individuals below the MCRS were caught, thus nP- was not estimated.

codends. No individuals below the MCRS of the species were caught by the two codends, therefore neither the *nP-* nor the discard ratios were estimated (Table 5). Regarding the Atlantic mackerel, no significant differences between codends were found concerning the number of individuals above the 15 cm MCRS retained (*nP+*; *CIs*: 22.8–152.2%). The few number of individuals below the MCRS caught by both codends did not allow to estimate both the *nP-* and the discard ratio indicators (Table 5).

A significant difference was found, for the Mediterranean scaldfish, concerning the number of individuals below the 12 cm reference size (*nP-*), of which DM50 retained, on average, the 33.7% (*CIs*: 8.1–94.0%) compared to SM40. On the contrary, we did not observe any significant difference, between codends, in both the *nP+* indicator (the *CIs* contained 100) and the discard ratios (the *CIs* of the two values overlapped; Table 5). Concerning the three species without a reference size, the percentages of all the individuals retained by DM50 compared to SM40 revealed a significant difference only for the spotted flounder. In fact, the average percentage caught by DM50 was around 26% (*CIs*: 18.7–35.2%) compared to SM40. Both the tub gurnard and the broadtail shortfin squid did not reveal any statistically significant difference in their catches between codends, since the *CIs* contained 100% indicating an equal number of individuals caught (Table 5).

## 4. Discussion

Mediterranean bottom trawl fisheries exert multiple impacts on the marine environment and ecosystem. Besides the physical alteration of the seabed [32–34] and the greenhouse gas emissions [35], these towed gears are well-known to cause a significant disturbance on the benthic habitats and communities [36, 37]. The present study is focused on evaluating the impact of a bottom trawl fishery from a species community perspective. To this aim, we performed a catch dominance analysis, to provide information on the species composition dynamics when

applying different technical modifications. This holistic approach, here investigated in the trawl selectivity field, allows to better understand and evaluate the impact of a fishery by including, in the analysis, not only those species having a commercial interest or requiring close protection or management measures, but all the species caught by the specific gear. In particular, we here compared the catches of the codends established for the Mediterranean, the SM40 codend, with those of the only alternative codend allowed after justified request, the DM50 codend [6].

The large number of species present in the catches of the two codends, which are usually not accounted in traditional selectivity studies, demonstrate the relevance of this methodology. Most of these species contribute to the discarded fraction of the catch, which accounts for the 50% (in weight) of all the catches obtained with both codends. This percentage falls within the range (20–65%) reported by Tsagarakis et al. [2] and gives a clear insight of the high impact exerted by this fishing gear on the benthic and benthopelagic community. The fate of the species entering the trawl codend is also quantified in the recent work of Mytilineou et al. [13], who estimated that less than 30% of the species caught were selected to be marketed. Also, in the present study, there were more species entirely discarded (37 species) than the ones possibly landed (31 species). The swimming crab *L. depurator*, which was by far the dominant animal species in the catches in both numbers and weight, is also a sign of the intense fishing effort carried out by trawlers in the Adriatic Sea, one of the most overexploited areas of Mediterranean [38, 39]. In fact, the intense discarding practices carried out by fishers in this area let the benthic opportunistic scavengers (i.e. some fish species, crabs as *L. depurator*, echinoderms and molluscs) consume these dead animals and thus thrive [40–42]. Some of these species, which can have a massive presence in bottom trawl catches, are starting to be marketed by some vessels in the study area (e.g. the largest individuals of *L. depurator*; Lucchetti, personal communication). This is clear evidence of the fishing down the marine food webs [43] as demonstrated by stock assessments in the Mediterranean, which highlight a situation of overexploitation for around 75% of the assessed commercial species [7].

The twaite shad, *A. fallax*, was the only protected species caught during the sea trials. In the Adriatic Sea, this species is commonly by-caught and discarded not only in demersal trawl fisheries [44] but also in pelagic trawl [45] and set net fisheries [46]. Another protected species, commonly subject to incidental captures from the same fishing gears, is the loggerhead sea turtle, *Caretta caretta* (listed in Annex IV of the Habitat Directive; [31]). Although no individuals were caught in the present study, bottom trawlers are responsible, in the Central-Northern Adriatic, for frequent bycatch events (around 8600 individuals per year; [47]).

Both the legal codends currently in use in the Mediterranean (SM40 and DM50) are known to be insufficiently size selective for many commercial species targeted by bottom trawl fisheries, since individuals below the MCRS and/or the length of first maturity are often retained in the catches [1]. Regarding the European hake, one of the most landed and overexploited demersal species in the whole region [48], the two codends were unable to avoid the catch of individuals either below the length of first maturity (more than 30 cm; [49, 50]) or below the MCRS of 20 cm. This is in line with several selectivity studies conducted on SM40 and DM50, where the predicted 50% retention length (L50) was always lower than 20 cm in different areas (Adriatic Sea [21, 51, 52]; Aegean Sea [53–56]; Alboran Sea [57]; Balearic Islands [58, 59] Tyrrhenian Sea [60]). Accordingly, in the present study, a discard ratio of 9–23% of the total catch in number of hakes was estimated. Interestingly, from the catch comparison results, we observed that SM40 caught significantly more undersized European hakes than DM50, contrary to what is usually expected (i.e. a similar size selectivity of the two codends for this species; see the review of Lucchetti et al. [1]). The exploitation pattern indicators confirm these results, stating that the DM50 was able to exclude from the catch, on average, half of the

undersized hakes observed in the SM40 catches. These findings are probably due to the slightly different mesh size (38.1 mm on average) of the SM40 codend compared with the Regulation requirements.

The same hypothesis could be done for the red mullet, another key target species in several Mediterranean fisheries, since the catch efficiency for the smaller length classes (11–14.5 cm) was significantly lower in the DM50 than in the SM40. The difference in the measured mesh size (around 12 mm) between DM50 and SM40 could have exerted, in the present study, a bigger effect on selectivity than the difference in the mesh configuration and/or mesh openness [61]. Nevertheless, the number of commercial individuals caught by the two codends was not significantly different, based on the exploitation pattern indicators analysis. However, the *CIs* for *nP+* were really wide (7.8–118.3%), reducing the power of these results. No red mullets below the 11 cm MCRS were caught in the sea trials. Although we cannot rule out that both codends avoided the catch of undersized individuals, since the experiment lacked the presence of a codend-cover to study the entire population entering the trawl net, Sala et al. [17] already demonstrated that the current legal codends provide a predicted L50 higher than the MCRS of the species. Accordingly, very few specimens of Atlantic mackerel under the 15 cm MCRS were caught by the two codends. Although there is no information on the size selectivity of SM40 and DM50 for this commercial species, Petetta et al. [62] estimated an L50 of more than 21 cm by using a 55 mm diamond mesh codend.

The significantly lower catch efficiency of DM50 compared to SM40, observed in both the catch comparisons and the exploitation pattern indicators, for the two flatfish species (Mediterranean scaldfish and spotted flounder) is in accordance with what found in the literature [51, 63, 64]. In fact, the flat morphology of these species fits better to the diamond-shaped meshes than to the square-shaped meshes [65]. That is the reason why, in trawl fisheries specifically targeting flatfish, such as the "Rapido" trawls in the Adriatic Sea [66], the advice is to only mount diamond mesh codends to increase the size selectivity [1]. The shift from one codend to another seemed not to influence the catch performance for tub gurnard and broadtail shortfin squid, but the wide *CIs* observed in both the catch comparison and exploitation pattern indicator analyses reflect a low number of individuals caught and do not allow to draw definitive conclusions.

The experimental design applied for the data collection in the present study was unpaired, with a first group of hauls carried out with one codend design and afterwards with the other design. Therefore, we cannot be 100% certain that the average populations entering the two codends were identical. However, since the hauls were carried out on the same cruise, within a relatively small geographical area and also within a limited time span, we assume that there only would be minor differences in the population size structures and species composition between the hauls conducted with SM40 and those with DM50. Therefore, we assume that the collected data can be used for a comparison between the performance of the two codends.

The avoidance of unwanted catches through improved selectivity is one of the primary goals for the implementation of the CFP in Mediterranean bottom trawl fisheries. In the last decades, several European projects (e.g. DISCATCH, DiscardLess, MINOUW, GALION, IMPLEMED) have focused on identifying measures, including technical ones related to fishing gear characteristics, to reduce discards and increase the fishers' awareness [8]. Furthermore, there is an increased interest of the scientific community towards the so-called LIFE (Low Impact and Fuel Efficient) fishing gears [67], which are more sustainable and could be employed as alternative gears. The methodology here described allows to evaluate more in-depth the overall impact of a fishing gear and to compare the catches obtained with two or more different gears from a species community perspective. Thus, it can definitely contribute to evaluate the economic and environmental viability of a specific fishing activity.

## Supporting information

**S1 Table. Dominance percentages of each species caught with the 50 mm diamond mesh codend (DM50) and the 40 mm square mesh codend (SM40), in both number of individuals and weight, for each fraction of the catch (total, discarded and landed).** The green, yellow, orange and red areas represent the target species, the bycatch species of commercial value, the species of no commercial value and the protected species, respectively.
(DOCX)

**S1 File. Data collected through the sea trials used for the analyses.** Data is divided into: haul data; catch data; frequency distribution data.
(XLSX)

## Acknowledgments

The research leading to these results has been conceived under the International PhD Program "Innovative Technologies and Sustainable Use of Mediterranean Sea Fishery and Biological Resources" (www.fishmed-phd.org). This study represents partial fulfilment of the requirements for the PhD thesis of A. Petetta. We are indebted to the crew of R/V "G. Dallaporta" for their valuable assistance in the field work.

## Author Contributions

**Conceptualization:** Andrea Petetta, Bent Herrmann, Alessandro Lucchetti.

**Data curation:** Andrea Petetta.

**Formal analysis:** Andrea Petetta, Bent Herrmann.

**Funding acquisition:** Alessandro Lucchetti.

**Investigation:** Andrea Petetta, Daniel Li Veli, Rocco De Marco, Alessandro Lucchetti.

**Methodology:** Andrea Petetta, Bent Herrmann, Daniel Li Veli.

**Project administration:** Massimo Virgili, Alessandro Lucchetti.

**Resources:** Massimo Virgili, Alessandro Lucchetti.

**Software:** Andrea Petetta, Bent Herrmann.

**Supervision:** Bent Herrmann, Massimo Virgili, Alessandro Lucchetti.

**Validation:** Andrea Petetta, Bent Herrmann, Daniel Li Veli, Massimo Virgili, Rocco De Marco, Alessandro Lucchetti.

**Writing – original draft:** Andrea Petetta.

**Writing – review & editing:** Andrea Petetta, Bent Herrmann, Daniel Li Veli, Massimo Virgili, Rocco De Marco, Alessandro Lucchetti.

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
