## [Decision Letter · Decision Letter 0]

2 Feb 2023

PONE-D-22-29329

Every animal matters! Evaluating the selectivity of a Mediterranean bottom trawl fishery from a species community perspective

PLOS ONE

Dear Dr. Petetta,

Thank you for submitting your manuscript to PLOS ONE. After careful consideration, we feel that it has merit but does not fully meet PLOS ONE’s publication criteria as it currently stands. Therefore, we invite you to submit a revised version of the manuscript that addresses the points raised during the review process.

Both reviewers agree that your work is interesting and worthy of eventual publication. Please consider comments from reviewers on discussing or highlighting further key points of your work. Reviewer #1 provides a useful list of such points that will improve your ms. Also kindly consider reducing the number of tables.

We look forward to receiving your revised manuscript.Kind regards,

Christos Maravelias, Ph.D.

Academic Editor

PLOS ONE

Journal Requirements:

2. In your Methods section, please provide additional information regarding the permits you obtained for the work. Please ensure you have included the full name of the authority that approved the field site access and, if no permits were required, a brief statement explaining why

Reviewers' comments:

Reviewer's Responses to Questions

**Comments to the Author**

1. Is the manuscript technically sound, and do the data support the conclusions?

Reviewer #1: Yes

Reviewer #2: Yes

2. Has the statistical analysis been performed appropriately and rigorously? 

Reviewer #1: Yes

Reviewer #2: Yes

3. Have the authors made all data underlying the findings in their manuscript fully available?

Reviewer #1: No

Reviewer #2: Yes

4. Is the manuscript presented in an intelligible fashion and written in standard English?

Reviewer #1: Yes

Reviewer #2: Yes

5. Review Comments to the Author

Reviewer #1: Dear Editor,

Please find my comments concerning the work entitled "Every animal matters! Evaluating the selectivity of a Mediterranean bottom trawl fishery from a species community perspective" (PONE-D-22-29329) below.

Sincerely yours,

General overview

The paper is a very interesting original contribution that presents important information from both a scientific and fisheries management point of view. The authors showed that a large amount of species of the catch in bottom trawl fishing, consisting of species without commercial value, which are discarded, are not commonly considered. This does not permit to correctly evaluate the ecological impact of this gear. The research meets all applicable standards for the ethics of experimentation. The use of dominance curves and the relevant plots was an ideal idea to show the objectives of this paper, combined with high technical standards in methods and analysis used in catch comparison studies, and by using specific exploitation indicators. The Results supported the conclusions. I would propose for the section of Discussion to be more developed. There are potential points that could be further discussed or highlighted (e.g. the low number of samples in some cases leading to weak or inconclusive results, the different hauls conducted for each tested gear and the potential consequences in the community and population structure, the lower than the legal 40 mm square mesh size used in the experimental fishing, the inefficiency of the trawl because of the low proportion of commercial species in the catch, a suggestion for the use of the DM50 for flatfish fishery management, the different than in the published literature results for hake and red mullet between the two tested codends, the similar discard ratio for the two codends, a suggestion to focus more on the study of the non-commercial part of the catch for future research). Nevertheless, it should be mentioned that I was really impressed by the simplicity and clearness of the text, which was easily readable even in the case that someone is not so familiar with the methods used. It is also worth mentioning that apart the usefulness of this paper for the Adriatic, it is a general paradigm for the Mediterranean bottom trawl research and fishery. Some specific comments are given below.

According to the above mentioned remarks, which I believe the authors can easily address, the paper could be accepted with minor changes.

Specific comments

Title:

well done

Abstract

I believe that the first 4 lines are too long as an introduction in this abstract. In addition, at the end of the abstract, something short is needed concerning the usability of the results in Mediterranean bottom trawl fisheries management and at regional level (e.g. the results for flatfish fishery).

Keywords

well done

Introduction

L59: you could mention here a recent paper of Mytilineou et al. (2022) concerning the same objectives (https://doi.org/10.3389/fmars.2022.1021467) , useful also for your discussion. See below the comments for Discussion.

L66: In reality, the SM40 codend made by meshes of 38.1 mm, as mentioned in L. 84 of M&M, is not legal. You may delete the word "legal" from here.

M&M

L84: Is this net officially used as a 40 mm square mesh net? As said before, this is not legal according to the EC Regulations. Could you explain this divergence? Why a 40 mm or more square mesh was not used if your objective was to compare the legal codends?

L85: was the circumference of the two codends similar? This factor can affect the selectivity.

L86-87: So, the hauls conducted with 40S and those with 50D were not the same in number and not conducted in the same date and grounds (are also shown in Table 2 and Fig. 1)? Could this produce a different community and population structure entering the trawl codend? This could also cause differences in the catch of the two codends. If so, please explain in Discussion, if this could also be a reason of differences between the two codends.

L91: delete "all"

L92-93: I suppose that the researchers divided "each fraction" by species (or the catch of each fraction).

L101: Chelidonichthys lucerna. Please check in WoRMS for Chelidonichthys lucernus (it is unacceptable). Then you should change the name in the text, tables and figures.

L137: ….summed up in the nominator.

L159: ..i.e. the ratios...

L167: subscript l in CCl

L171: 32 or 31?

L213: ...to a MCRS....

Results

Table 3: I was wondering about the absence of economic value for species, such as P. acarne, P. bogaraveo, P. erythrinus, P. pagrus, E. gurnardus, S. sphyraena and B. boops, which is not the case in other areas of Mediterranean; similarly, about the bycatch character of P. longirostris although it is commonly a target species or that of A. laterna, which is discarded elsewhere. Is this phenomenon related to local consumers’ preferences?

L254: I believe that Table 4 can be presented as supplementary material since Fig 2 & 3 show the main results.

L260: There is no need for a paragraph here. The text can be merged to one paragraph with the previous one (L259).

L261-262: delete the parentheses

L273: Again, there is no need for a paragraph here, which is also small. The text can be merged to one paragraph with the previous one (L272).

L278: again, merge with the previous paragraph (L277).

L283-285: I think, the expression that "...DM50 caught a significant larger proportion of the species S10-S17...." is not correct in the case of landings. You may use "produced" instead of "caught", because this may be the result of gear and fisher selection too.

L288-292: these differences are statistically significant, however, are they important? They seem to present a low % difference and the CIs are close to the axe 0.0. In addition they represent the last asymptotic part of the cumulative curve.

Fig. 3: The plot for the total catch in terms of weight shows no differences for the first 31 species (except S2) between the two codends. However, there are important differences in terms of numbers. This may indicate smaller individuals included in the catch of SM40 than in DM50. However, I am wondering if these differences between the two codends are really a result of different catch efficiency or are related more to a different structure of the entering populations in each codend. This may be a factor affecting the dominance curves. You may want to discuss this point.

L295-323: The results concerning Fig. 4 could be presented in a shorter way by avoiding many details and focusing more on the main differences, useful to explain differences in the cumulative dominance curves between the two codends.

L326-328: change to “Table 5 reports the number of individuals measured in each haul for the 7 species selected for the catch comparison analysis (M. merluccius, M. barbatus, S. scombrus, A. laterna, C. linguatula, I. coindetii, C. lucernus).”

L336: The first sentence (Fig. 5….basin) could be deleted to avoid duplication, and the rest of the paragraph could be merged with the previous paragraph.

L353: again, you could merge this paragraph with the previous one.

Table 7: The CIs are very wide in many cases, which may make some results less powerful. This may be related to the low number of individuals in your samples. You could comment on this in Discussion.

Discussion

L457: again, you should avoid the term legal since the mesh size of your SM40 is lower than 40 mm.

L463: ….benthic and benthopelagic community.

L468: (….crabs such as L. depurator, echinoderms..)

L461-464: It would be interesting to compare your results with those of the work of Mytilineou et al. (2022).

L485: add references showing that the selectivity of the examined gears is lower than the MCRS of the species. There are a lot on this topic.

L490-491: As mentioned before, another reason explaining these results may be the difference in the population structure of the species in the sea (which could be possible in this case since the hauls conducted for each gear were different). If the population fished by SM40 contained more juvenile hake than that of DM50, then the catch of SM40 will possibly contain more undersized hake than the DM50, which will produce a high value for nP-; similarly for nP+. The important result here is that the nDRatioT and nDRatioB do not differ, which indicate that the proportion of discarded individuals to the total amount caught is similar in the two codends. I would expect a discussion on this.

Tables

Table 7: in the legend of this table, add information concerning the gears studied.

Reviewer #2: This manuscript addresses the catch performance and selectivity of Mediterranean bottom trawl fishery by using a broader approach that considers the entire species community affected by trawling. I do agree with the authors that standard selectivity studies have focused on few commercially important and/or the most vulnerable species in the fishery, while neglecting large fractions of the catch. I do agree also that this procedure for selectivity studies results in underestimation of unaccounted fishing mortality when evaluating the ecological impact of using the specific fishing gear. I find this well-written manuscript very interesting and innovative. It expands the horizon of research in the field of fishing gear technology and puts selectivity in an ecosystem-oriented perspective. I recommend this manuscript to be published by the journal after authors address some few minor comments.

1. FALL IN SCOPE WITH JOURNAL

Subject fall well within the scope of the journal

2. NEW AND ORIGINAL

Introducing assessment of biodiversity and combining this with catch comparisons and exploitation patters is a new, broader way of assessing catch efficiency of fishing gears.

3. TITLE AND CONTENTS

Ok, it reflexes what is presented in the manuscript.

4. ABSTRACT

Ok.

5. KEYWORDS.

Ok.

6. INTRODUCTION

Ok

line 50-52: should specify at you are taking of fullmesh size.

7. OBJECTIVES

The objective is OK and sensible.

8. MATERIALS AND METHODS

The set of experiments is well planned, the collection of data is appropriate and the analysis is valid.

line 77: should be "2-panel" net

9. RESULTS

Ok.

10. DISCUSSION

Ok.

11. TABLES AND FIGURES

The manuscript has too many tables. I would suggest removing table 2 and give the information in the text. For example: hauls 1-9 were done with SM40, mean haul duration (SD) was 60 min (3 min), mean depth (SD) was 54.3 m (2m)... hauls 10-20 were done with DM50....

Was there any difference in depth, haul duration or horizontal net opening? if yes, was this data used in the analysis? if yes, how?

12. REFERENCES

Ok

13. ENGLISH

Ok

6. PLOS authors have the option to publish the peer review history of their article (what does this mean?). If published, this will include your full peer review and any attached files.

Reviewer #1: No

Reviewer #2: No

---

## [Author Response · Author response to Decision Letter 0]

17 Feb 2023

Revision note for Manuscript Number PONE-D-22-29329:

“Every animal matters! Evaluating the selectivity of a Mediterranean bottom trawl fishery from a species community perspective”

Editor

Comment 1

Thank you for submitting your manuscript to PLOS ONE. After careful consideration, we feel that it has merit but does not fully meet PLOS ONE’s publication criteria as it currently stands. Therefore, we invite you to submit a revised version of the manuscript that addresses the points raised during the review process.

Response 1

We are pleased that the Editor invites us to resubmit the manuscript after revision which we hereby do.

Comment 2

Both reviewers agree that your work is interesting and worthy of eventual publication. Please consider comments from reviewers on discussing or highlighting further key points of your work. Reviewer #1 provides a useful list of such points that will improve your ms. Also kindly consider reducing the number of tables.

Response 2

We have addressed all the reviewers’ comments and suggestions below. The number of tables has been reduced from 7 to 5, since Table 2 was deleted and Table 3 was moved to Supplementary Information.

 

Reviewer 1: 

General overview

Comment 3

The paper is a very interesting original contribution that presents important information from both a scientific and fisheries management point of view. The authors showed that a large amount of species of the catch in bottom trawl fishing, consisting of species without commercial value, which are discarded, are not commonly considered. This does not permit to correctly evaluate the ecological impact of this gear. The research meets all applicable standards for the ethics of experimentation. The use of dominance curves and the relevant plots was an ideal idea to show the objectives of this paper, combined with high technical standards in methods and analysis used in catch comparison studies, and by using specific exploitation indicators. The Results supported the conclusions. I would propose for the section of Discussion to be more developed. There are potential points that could be further discussed or highlighted (e.g. the low number of samples in some cases leading to weak or inconclusive results, the different hauls conducted for each tested gear and the potential consequences in the community and population structure, the lower than the legal 40 mm square mesh size used in the experimental fishing, the inefficiency of the trawl because of the low proportion of commercial species in the catch, a suggestion for the use of the DM50 for flatfish fishery management, the different than in the published literature results for hake and red mullet between the two tested codends, the similar discard ratio for the two codends, a suggestion to focus more on the study of the non-commercial part of the catch for future research). Nevertheless, it should be mentioned that I was really impressed by the simplicity and clearness of the text, which was easily readable even in the case that someone is not so familiar with the methods used. It is also worth mentioning that apart the usefulness of this paper for the Adriatic, it is a general paradigm for the Mediterranean bottom trawl research and fishery. Some specific comments are given below.

According to the above mentioned remarks, which I believe the authors can easily address, the paper could be accepted with minor changes.

Response 3

We thank the reviewer for the suggestions that helped us to significantly improve the MS; we have addressed each specific comment below.

Specific comments

Title:

well done

Abstract

Comment 4

I believe that the first 4 lines are too long as an introduction in this abstract. In addition, at the end of the abstract, something short is needed concerning the usability of the results in Mediterranean bottom trawl fisheries management and at regional level (e.g. the results for flatfish fishery).

Response 4

According to this suggestion, we deleted the following sentence in the abstract: “This can lead to neglecting large fractions of the catch and to underestimate the unaccounted fishing mortality when evaluating the ecological impact of using the specific fishing gear.” Moreover, a sentence was added at the end of the abstract: “The outcomes of the study can be useful for the Mediterranean bottom trawl fisheries management.”

Keywords

well done

Introduction

Comment 5

L59: you could mention here a recent paper of Mytilineou et al. (2022) concerning the same objectives (https://doi.org/10.3389/fmars.2022.1021467), useful also for your discussion. See below the comments for Discussion.

Response 5

We thank the reviewer for this suggestion and updated the sentence as follows: “On the contrary, the multispecies nature of Mediterranean bottom trawl fisheries is rarely addressed, despite the growing need for ecosystem-based fishery management for conserving biodiversity (Mytilineou et al., 2022).”

Comment 6

L66: In reality, the SM40 codend made by meshes of 38.1 mm, as mentioned in L. 84 of M&M, is not legal. You may delete the word "legal" from here.

Response 6

We agree with the reviewer and we deleted the word “legal”.

M&M

Comment 7

L84: Is this net officially used as a 40 mm square mesh net? As said before, this is not legal according to the EC Regulations. Could you explain this divergence? Why a 40 mm or more square mesh was not used if your objective was to compare the legal codends?

Response 7

The reviewer is right. The nominal mesh size of this codend is 40 mm, but the 20 measurements with ICES omega gauge revealed that it was significantly less than 40 mm in measured mesh size. Obviously, we would have liked that the actual mesh size was according to legislation, however it was what it was and we need to report the actual. Further, to be consistent with the other measures given in the MS, we shifted from the standard error to the standard deviation also for measured mesh size (38.11 ± 0.94; see Table 1 revised).

Comment 8

L85: was the circumference of the two codends similar? This factor can affect the selectivity.

Response 8

The circumference of the two codends was similar. To better clarify this on the MS, we wrote the following sentence: “The number of meshes at codend circumference was 140 for SM40 and 220 for DM50 (Table 1), to obtain a similar circumference between codends during fishing, thus avoiding different effects on selectivity. In fact, following the considerations and calculations made by Sala et al. [17] on the expected mesh openness during fishing in square and diamond configurations, respectively, the expected circumference of both codends were around 2.7 meters.”

Comment 9

L86-87: So, the hauls conducted with 40S and those with 50D were not the same in number and not conducted in the same date and grounds (are also shown in Table 2 and Fig. 1)? Could this produce a different community and population structure entering the trawl codend? This could also cause differences in the catch of the two codends. If so, please explain in Discussion, if this could also be a reason of differences between the two codends.

Response 9

The number of hauls carried out with SM40 and DM50 was not identical, however this did not affect the catch comparison, catch ratio and catch dominance estimates. We added the following text in the Discussion section: “The experimental design applied for the data collection in the present study was unpaired, with a first group of hauls carried out with one codend design and afterwards with the other design. Therefore, we cannot be 100% certain that the average populations entering the two codends were identical. However, since the hauls were carried out on the same cruise, within a relatively small geographical area and also within a limited time span, we assume that there only would be minor differences in the population size structures and species composition between the hauls conducted with SM40 and those with DM50. Therefore, we assume that the collected data can be used for a comparison between the performance of the two codends.”

Comment 10

L91: delete "all"

Response 10

Deleted.

Comment 11

L92-93: I suppose that the researchers divided "each fraction" by species (or the catch of each fraction).

Response 11

According to this suggestion, the sentence was rewritten as follows: “Then, for each fraction, the researchers on-board divided the catch of each fraction by species…”

Comment 12

L101: Chelidonichthys lucerna. Please check in WoRMS for Chelidonichthys lucernus (it is unacceptable). Then you should change the name in the text, tables and figures.

Response 12

We thank the reviewer for this comment and we corrected the name accordingly throughout the MS.

Comment 13

L137: ….summed up in the nominator.

Response 13

Corrected.

Comment 14

L159: ..i.e. the ratios...

Response 14

Corrected.

Comment 15

L167: subscript l in CCl

Response 15

Done.

Comment 16

L171: 32 or 31?

Response 16

The sentence was rewritten as follows: “We estimated the catch comparison rate, among these models…”

Comment 17

L213: ...to a MCRS....

Response 17

Corrected.

Results

Comment 18

Table 3: I was wondering about the absence of economic value for species, such as P. acarne, P. bogaraveo, P. erythrinus, P. pagrus, E. gurnardus, S. sphyraena and B. boops, which is not the case in other areas of Mediterranean; similarly, about the bycatch character of P. longirostris although it is commonly a target species or that of A. laterna, which is discarded elsewhere. Is this phenomenon related to local consumers’ preferences?

Response 18

Yes, the classification was made following the fishers’ choice at the time of the experiments. To make it clearer in the MS, we added the following sentence: “This classification was specifically related to the fishers’ choice in a precise spatio-temporal context. Therefore, it does not fully represent neither all the Mediterranean bottom trawl fisheries, where some species, here classified as commercial, could be always discarded elsewhere and vice versa due to local consumers’ preferences, nor the Adriatic fishery, where some species can have a market value only in a specific season.”

Comment 19

L254: I believe that Table 4 can be presented as supplementary material since Fig 2 & 3 show the main results.

Response 19

Done.

Comment 20

L260: There is no need for a paragraph here. The text can be merged to one paragraph with the previous one (L259).

Response 20

Corrected.

Comment 21

L261-262: delete the parentheses

Response 21

Done.

Comment 22

L273: Again, there is no need for a paragraph here, which is also small. The text can be merged to one paragraph with the previous one (L272).

Response 22

Corrected.

Comment 23

L278: again, merge with the previous paragraph (L277).

Response 23

Corrected.

Comment 24

L283-285: I think, the expression that "...DM50 caught a significant larger proportion of the species S10-S17...." is not correct in the case of landings. You may use "produced" instead of "caught", because this may be the result of gear and fisher selection too.

Response 24

We agree with the reviewer and corrected accordingly.

Comment 25

L288-292: these differences are statistically significant, however, are they important? They seem to present a low % difference and the CIs are close to the axe 0.0. In addition they represent the last asymptotic part of the cumulative curve.

Response 25

According to this suggestions, these lines were rewritten as follows: “The same trend is observed, in the discarded catch, for species S1 to S31 i.e. all the species of a commercial value, especially from S18 to S31. Other barely significant differences (the upper CI almost reaches the 0.0 line) are observed in the left end of the curves of both total and discarded fractions from species S45 to S58 i.e. ‘Species of no commercial value’, whose proportion in the DM50 codend is slightly less than in the SM40 codend.”

Comment 26

Fig. 3: The plot for the total catch in terms of weight shows no differences for the first 31 species (except S2) between the two codends. However, there are important differences in terms of numbers. This may indicate smaller individuals included in the catch of SM40 than in DM50. However, I am wondering if these differences between the two codends are really a result of different catch efficiency or are related more to a different structure of the entering populations in each codend. This may be a factor affecting the dominance curves. You may want to discuss this point.

Response 26

We have addressed the reviewer’s comment before. Please see Response 9.

Comment 27

L295-323: The results concerning Fig. 4 could be presented in a shorter way by avoiding many details and focusing more on the main differences, useful to explain differences in the cumulative dominance curves between the two codends.

Response 27

We agree with this reviewer’s suggestion, and the paragraph was rewritten as follows:

“Fig. 4 shows the delta plots resulting from the comparison of the catch dominance curves, for the total, discarded and landed fractions of the catch. This figure provides a detailed insight at each single species level. The plots of total and discarded fractions show, in particular, a significant difference between codends for the European hake (S1) and the two flatfish species (S18, S19), indicating a larger proportion of their dominance in the SM40 catches than in the DM50 catches in both number of individuals (Fig. 4, left column) and weight (Fig. 4, right column). The delta plot of the total catch also shows a significantly larger proportion of red mullet (S2) in SM40 than in DM50 catches, only in weight; on the contrary, a significantly larger proportion of monkfish (S3) is present in DM50 catches, in both number of individuals and weight. Also, L. depurator (S32) had a significantly larger proportion in the total catch of DM50 compared to SM40, only in number of individuals. Other differences detected at species level for these two fractions are barely significant. The delta plot of the landed fraction, in both number of individuals and weight, shows in particular that a larger proportion of the target species M. merluccius (S1), Lophius spp. (S3), the cephalopods S. officinalis (S10) and Eledone spp. (S11), is present in the DM50 catches than in the SM40 catches. On the contrary, a larger proportion of M. barbatus (S2), T. minutus capelanus (S5) and C. linguatula (S18) is present in SM40 catches than in DM50 catches.”

Comment 28

L326-328: change to “Table 5 reports the number of individuals measured in each haul for the 7 species selected for the catch comparison analysis (M. merluccius, M. barbatus, S. scombrus, A. laterna, C. linguatula, I. coindetii, C. lucernus).”

Response 28

Corrected.

Comment 29

L336: The first sentence (Fig. 5….basin) could be deleted to avoid duplication, and the rest of the paragraph could be merged with the previous paragraph.

Response 29

Corrected.

Comment 30

L353: again, you could merge this paragraph with the previous one.

Response 30

Corrected.

Comment 31

Table 7: The CIs are very wide in many cases, which may make some results less powerful. This may be related to the low number of individuals in your samples. You could comment on this in Discussion.

Response 31

According to this suggestion, two sentences have been added in the Discussion section. The first concerned the red mullet: “Nevertheless, the number of commercial individuals caught by the two codends was not significantly different, based on the exploitation pattern indicators analysis. However, the CIs for nP+ were really wide (7.8-118.3%), reducing the power of these results.”

Another sentence concerned the tub gurnard and squid: “The shift from one codend to another seemed not to influence the catch performance for tub gurnard and broadtail shortfin squid, but the wide CIs observed in both the catch comparison and exploitation pattern indicator analyses reflect a low number of individuals caught and do not allow to draw definitive conclusions.”

Discussion

Comment 32

L457: again, you should avoid the term legal since the mesh size of your SM40 is lower than 40 mm.

Response 32

According to this and the previous suggestions, we deleted the term “legal” from this line.

Comment 33

L463: ….benthic and benthopelagic community.

Response 33

Corrected.

Comment 34

L468: (….crabs such as L. depurator, echinoderms..)

Response 34

Done.

Comment 35

L461-464: It would be interesting to compare your results with those of the work of Mytilineou et al. (2022).

Response 35

According to this suggestion, we added a sentence as follows: “The fate of the species entering the trawl codend is also described in the recent work of Mytilineou et al. [13], who estimated that less than 30% of the species caught were selected to be marketed.”

Comment 36

L485: add references showing that the selectivity of the examined gears is lower than the MCRS of the species. There are a lot on this topic.

Response 36

According to this suggestion, we added a sentence as follows: “This is in line with several selectivity studies conducted on SM40 and DM50, where the predicted 50% retention length (L50) was always lower than 20 cm in different areas (Adriatic Sea [21,51,52]; Aegean Sea [53–56]; Alboran Sea [57]; Balearic Islands [58,59] Tyrrhenian Sea [60]).” 

Comment 37

L490-491: As mentioned before, another reason explaining these results may be the difference in the population structure of the species in the sea (which could be possible in this case since the hauls conducted for each gear were different). If the population fished by SM40 contained more juvenile hake than that of DM50, then the catch of SM40 will possibly contain more undersized hake than the DM50, which will produce a high value for nP-; similarly for nP+. The important result here is that the nDRatioT and nDRatioB do not differ, which indicate that the proportion of discarded individuals to the total amount caught is similar in the two codends. I would expect a discussion on this.

Response 37

We refer the reviewer to Response 9, where this point has been addressed.

Tables

Comment 38

Table 7: in the legend of this table, add information concerning the gears studied.

Response 38

According to this suggestion, the legend was updated as follows: “Table 7. Values of the exploitation pattern indicators (in average percentages with 95% confidence intervals) for the species selected for the catch comparison analysis. They represent the number of individuals i.e. nP (total), nP- (below the reference size), nP+ (above the reference size) retained by the test codend (DM50) compared to the baseline codend (SM40), and the resulting discard ratios estimated for both the test (nDRatioT) and baseline (nDRatioB) codends.”

 

Reviewer #2

Comment 39

This manuscript addresses the catch performance and selectivity of Mediterranean bottom trawl fishery by using a broader approach that considers the entire species community affected by trawling. I do agree with the authors that standard selectivity studies have focused on few commercially important and/or the most vulnerable species in the fishery, while neglecting large fractions of the catch. I do agree also that this procedure for selectivity studies results in underestimation of unaccounted fishing mortality when evaluating the ecological impact of using the specific fishing gear. I find this well-written manuscript very interesting and innovative. It expands the horizon of research in the field of fishing gear technology and puts selectivity in an ecosystem-oriented perspective. I recommend this manuscript to be published by the journal after authors address some few minor comments.

Response 39

We thank the reviewer for the comment, and we have addressed the specific comments below.

1. FALL IN SCOPE WITH JOURNAL

Subject fall well within the scope of the journal

2. NEW AND ORIGINAL

Introducing assessment of biodiversity and combining this with catch comparisons and exploitation patters is a new, broader way of assessing catch efficiency of fishing gears.

3. TITLE AND CONTENTS

Ok, it reflexes what is presented in the manuscript.

4. ABSTRACT

Ok.

5. KEYWORDS.

Ok.

6. INTRODUCTION

Ok

Comment 40

line 50-52: should specify at you are taking of fullmesh size.

Response 40

Done.

7. OBJECTIVES

The objective is OK and sensible.

8. MATERIALS AND METHODS

The set of experiments is well planned, the collection of data is appropriate and the analysis is valid.

Comment 41

line 77: should be "2-panel" net

Response 41

Corrected.

9. RESULTS

Ok.

10. DISCUSSION

Ok.

11. TABLES AND FIGURES

Comment 42

The manuscript has too many tables. I would suggest removing table 2 and give the information in the text. For example: hauls 1-9 were done with SM40, mean haul duration (SD) was 60 min (3 min), mean depth (SD) was 54.3 m (2m)... hauls 10-20 were done with DM50....

Response 42

According to this suggestion, Table 2 was removed. The paragraph was rewritten as follows: “Each of the two codends was mounted on the same trawl. 19 valid hauls were performed: the first 9 hauls with SM40 and the last 10 with DM50 (Table 2). All hauls were carried out in daylight at a mean depth (SD) of 55.5 meters (1.2), with a standardized towing duration of around 60 minutes (59.2 ± 1.9). The average towing speed was 3.0 knots (range 2.8-3.2 knots). The horizontal net opening (19.5 ± 0.2 meters) was monitored using acoustic sensors (SIMRAD, Norway).”

Also, according to the suggestions of reviewer 1, Table 4 was moved as supplementary information.

Comment 43

Was there any difference in depth, haul duration or horizontal net opening? if yes, was this data used in the analysis? if yes, how?

Response 43

There were no significant differences in both the mean haul duration, mean depth and mean horizontal net opening between the two codends tested; therefore, this data was not used in the analysis.

12. REFERENCES

Ok

13. ENGLISH

Ok

Kind regards,

Andrea Petetta, on behalf of all co-authors.

---

## [Editor Report · Decision Letter 1]

7 Mar 2023

Every animal matters! Evaluating the selectivity of a Mediterranean bottom trawl fishery from a species community perspective

PONE-D-22-29329R1

Dear Dr. Petetta,

We’re pleased to inform you that your manuscript has been judged scientifically suitable for publication and will be formally accepted for publication once it meets all outstanding technical requirements.

Kind regards,

Christos Maravelias, Ph.D.

Academic Editor

PLOS ONE
---

## [Editor Report · Acceptance letter]

12 Mar 2023

PONE-D-22-29329R1 

Every animal matters! Evaluating the selectivity of a Mediterranean bottom trawl fishery from a species community perspective 

Dear Dr. Petetta:

I'm pleased to inform you that your manuscript has been deemed suitable for publication in PLOS ONE. Congratulations! Your manuscript is now with our production department. 

Kind regards, 

on behalf of

Dr. Christos Maravelias 

Academic Editor

PLOS ONE